# Exploration of CTCF post-translation modifications uncovers Serine-224 phosphorylation by PLK1 at pericentric regions during the G2/M transition

**Brian C Del Rosario[1,2†], Andrea J Kriz[1,2†], Amanda M Del Rosario[3], Anthony Anselmo[4], Christopher J Fry[5], Forest M White[3], Ruslan I Sadreyev[4], Jeannie T Lee[1,2*]**

[1]Department of Molecular Biology, Howard Hughes Medical Institute, Massachusetts General Hospital, Boston, United States; [2]Department of Genetics, Harvard Medical School, Boston, United States; [3]Koch Institute for Integrative Cancer Research, Massachusetts Institute of Technology, Cambridge, United States; [4]Department of Molecular Biology, Massachusetts General Hospital, Boston, United States; [5]Cell Signaling Technology, Danvers, United States

**\*For correspondence:**
lee@molbio.mgh.harvard.edu

[†]These authors contributed equally to this work

**Abstract** The zinc finger CCCTC-binding protein (CTCF) carries out many functions in the cell. Although previous studies sought to explain CTCF multivalency based on sequence composition of binding sites, few examined how CTCF post-translational modification (PTM) could contribute to function. Here, we performed CTCF mass spectrometry, identified a novel phosphorylation site at Serine 224 (Ser[224]-P), and demonstrate that phosphorylation is carried out by Polo-like kinase 1 (PLK1). CTCF Ser[224]-P is chromatin-associated, mapping to at least a subset of known CTCF sites. CTCF Ser[224]-P accumulates during the G2/M transition of the cell cycle and is enriched at pericentric regions. The phospho-obviation mutant, S224A, appeared normal. However, the phospho-mimic mutant, S224E, is detrimental to mouse embryonic stem cell colonies. While ploidy and chromatin architecture appear unaffected, S224E mutants differentially express hundreds of genes, including p53 and p21. We have thus identified a new CTCF PTM and provided evidence of biological function.
DOI: https://doi.org/10.7554/eLife.42341.001

## Introduction

CCCTC-binding protein (CTCF) has been studied in many capacities since its discovery over twenty years ago. Originally identified as a candidate transcription regulator of *c-myc*, this multi zinc finger protein is highly conserved amongst Nephrozoa (*Heger et al., 2012*; *Lobanenkov et al., 1990*). Subsequent studies further revealed that CTCF has insulatory activity, specifically in blocking enhancer-promoter interactions, and in separating transcriptionally active genomic regions from heterochromatic domains (*Merkenschlager and Odom, 2013*; *Ong and Corces, 2014*). Later genomic studies revealed tens of thousands of CTCF binding sites in mammalian genomes (*Kim et al., 2007*; *Rhee and Pugh, 2011*; *Nakahashi et al., 2013*; *Schmidt et al., 2012*; *Xie et al., 2007*), with >5000 of these being conserved (*Schmidt et al., 2012*). In addition to CTCF's role in blocking enhancer-promoter loops, recent chromatin conformation capture (3C) based assays revealed that CTCF paradoxically also plays an architectural role in shaping the genome as well, helping to mediate three-dimensional chromatin loops in some cases. In addition, 3C-based assays also revealed that CTCF binds to the borders of many Topologically Associating Domains (TADs), megabase-sized regions

which function to insulate chromatin interactions such as promoter-enhancer loops (*Ong and Corces, 2014*; *Dixon et al., 2012*; *Nora et al., 2012*). Disruption of TADs, for example through deletion of CTCF sites, can lead to genetic disease through ectopic enhancer-mediated upregulation of genes (*Lupiáñez et al., 2015*; *Rao et al., 2014*; *Sanborn et al., 2015*; *Nora et al., 2017*).

How can a single protein carry out so many different functions in the cell? In addition, how does the cell specify which function CTCF carries out at a given binding site, especially when some of these functions, for example mediating versus blocking chromatin loops, directly contradict each other? Numerous studies have attempted to explain the multivalency of CTCF on the basis of factors such as motif composition of binding site, chromatin state, DNA methylation, and CTCF's role in mediating three-dimensional chromatin interactions (*Lu et al., 2016*). For example, it has been proposed that CTCF's insulatory activity can be explained in context of its mediation of three-dimensional chromatin loops.

In addition to these models, which predict the function of CTCF based on the chromatin state around a given binding site, other studies have sought to define CTCF function based on post-translation modifications (PTMs) of the protein itself. Poly(ADP-ribosyl)ation of CTCF has been found to play roles in imprinting and nucleolar transcription (*Torrano et al., 2006*; *Yu et al., 2004*) while SUMOylation of CTCF appears to enhance its repressive function (*MacPherson et al., 2009*). Finally, phosphorylation of CTCF has been proposed to turn CTCF into a transcription activator (*El-Kady and Klenova, 2005*) or reduce its DNA-binding activity (*Sekiya et al., 2017*), depending on site. These prior studies demonstrate that CTCF PTMs, possibly in combination, could influence the myriad of CTCF functions. The extent to which CTCF is post-translationally modified is currently no known. Many prior PTMs were identified indirectly by candidate-based approaches. Several proteomic screens have been carried out using mass spectrometry (*Rigbolt et al., 2011*; *Olsen et al., 2010*; *Kettenbach et al., 2011*; *Dephoure et al., 2008*), though phosphorylation sites were called by confidence scores made by probabilistic scoring algorithms without further validation or characterization. Here, we endeavored to screen for new CTCF PTMs by taking an unbiased approach. We identify a novel phosphorylation site at Serine 224 (Ser$^{224}$-P), perform an extensive characterization of its cellular functions, and uncover an effect on growth of embryonic stem cell colonies and gene regulation.

## Results

### Murine CTCF is phosphorylated at a highly conserved position, Ser$^{224}$

To explore the possibility that PTMs regulate CTCF, we took an unbiased approach to both confirm known and identify novel PTMs, specifically S/T/Y phosphorylation using immunoprecipitation and mass spectrometry (*Sekiya et al., 2017*; *Rigbolt et al., 2011*; *Olsen et al., 2010*; *Kettenbach et al., 2011*; *Dephoure et al., 2008*; *Klenova et al., 2001*). We utilized a doxycycline-inducible system to express murine CTCF-3xFLAG (*Sun et al., 2013*). This allowed us to purify CTCF from cells using a FLAG epitope rather than a CTCF antibody, as the latter could select against PTMs present within the antigenic sequence. The CTCF-3xFLAG transgene was stably introduced into female immortalized rtTA MEFs and doxycycline induction was verified by GFP microscopy (*Figure 1A*) (*Jeon and Lee, 2011*). Anti-FLAG immunofluorescence confirmed that CTCF-3xFLAG co-stained the nucleus in the same pattern as total CTCF (*Figure 1B*). Exogenous expression of CTCF-3xFLAG in this system only occurs at a modest level, which we confirmed by western blot (*Figure 1C*) (*Sun et al., 2013*). Previously, we also found that even slight perturbation of CTCF levels affects differentiation of female ES cells (*Sun et al., 2013*). However, induction of the transgene was well-tolerated in SV40T immortalized MEFs as they were viable and stably expressing CTCF-3xFLAG even after 9 days (*Figure 1C*). We thus concluded that exogenously expressed CTCF-3xFLAG in immortalized MEFs could be a tractable system for analyzing CTCF PTMs.

We expanded the CTCF-3xFLAG MEFs with or without induction for 3 days and nuclei were isolated. CTCF is tightly associated with chromatin through the central zinc fingers binding its core DNA motif (*Nakahashi et al., 2013*; *Yusufzai and Felsenfeld, 2004*). However, homogenizing nuclei under mild conditions with nucleases was not preferable as endogenous phosphatase activity would have been present. Thus we generated nuclear extracts under stringent conditions. Following dialysis of the nuclear extracts, anti-FLAG immunoprecipitation was performed and eluted material was

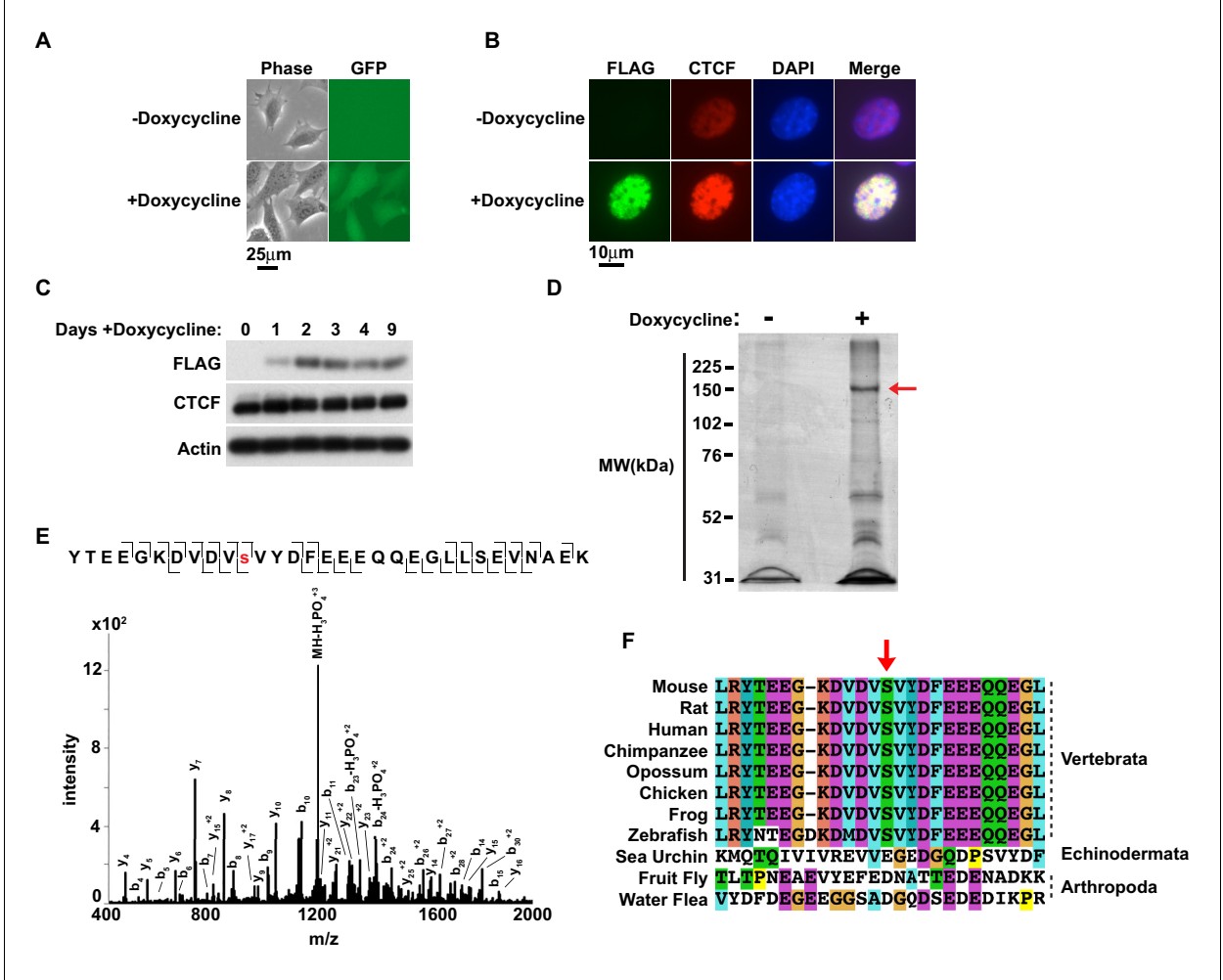

**Figure 1.** Murine CTCF is phosphorylated at Ser[224]. (**A**) GFP microscopy of co-inducible CTCF-3xFLAG, EGFP MEFs ± 1 μg/mL doxycycline for 48 hr. Bar, 25 μm. (**B**) FLAG and CTCF immunofluorescence accompanying (**A**). Bar, 10 μm. (**C**) FLAG, CTCF, and β-actin western blot of whole cell extracts from inducible CTCF-3XFLAG MEFs treated up to 9 days with 1 μg/mL doxycycline. (**D**) Coomassie stained gel of anti-FLAG immunoprecipitate from CTCF-3xFLAG MEFs treated 72hrs ± 1μg/mL doxycycline. Red arrow, band analyzed by mass spectrometry. (**E**) Manually validated mass spectra of CTCF peptide Tyr[214]-Lys[244] with y and b ions identified. Phosphorylation event at Ser[224] indicated by red s. (**F**) ClustalX alignment of CTCF sequences from the indicated species. Shown is a 25 amino acid window centered on mouse Ser[224] (red arrow).

DOI: https://doi.org/10.7554/eLife.42341.002

The following figure supplement is available for figure 1:

**Figure supplement 1.** Casein kinase 2 (CK2) can phosphorylate CTCF Ser[224].

DOI: https://doi.org/10.7554/eLife.42341.003

resolved by SDS-PAGE and Coomassie Blue stained. A band corresponding to CTCF-3xFLAG was excised for mass spectrometry analysis (arrow, *Figure 1D*). Following trypsin digestion and extraction, phosphorylated peptides were enriched using immobilized metal affinity chromatography (IMAC) with Fe-NTA resin prior to LC MS/MS. A unique murine site of serine phosphorylation was identified at Ser[224], which is ~50 aa N-terminal of the CTCF zinc finger domain (*Figure 1E*). Notably, in a phosphoproteome screen of HeLa cells, a single peptide containing human CTCF Ser[224]-P was identified (*Kettenbach et al., 2011*). However, the significance of this PTM in human cells was neither validated nor explored. While it is also reported that CTCF is phosphorylated at Ser[604], Ser[609], Ser[610] and Ser[612] (*El-Kady and Klenova, 2005*; *Rigbolt et al., 2011*; *Olsen et al., 2010*; *Dephoure et al., 2008*; *Klenova et al., 2001*); we were unable to identify peptides containing these residues. As these phospho-sites lie in close proximity in a region of CTCF poorly cut by trypsin and chymotrypsin, it is possible that resolution of these phospho-sites was infeasible with our approach.

Previous mass spectrometry studies which identified these residues were also global studies that did not provide spectra or additional validation of sites, making direct comparison with our study difficult. Likewise, we were also unable to confirm if the linker sequences between the zinc fingers were phosphorylated as previously reported (*Sekiya et al., 2017*).

Interestingly, CTCF orthologs have only been found in Nephrozoa (*Heger et al., 2012*). To explore the potential evolutionary significance of CTCF Ser[224]-P, we compared CTCF amino acid sequences from several Nephrozoa species with Clustal (*Figure 1F*). The CTCF amino acid sequences from mouse, rat, human, chimpanzee, opossum, chicken, frog, zebrafish, sea urchin, fruit fly, and water flea were aligned. A 25 amino acid window centered on mouse Ser[224] remarkably revealed striking conservation proximal to this position amongst vertebrata, but not echinodermata or arthropoda (arrow, *Figure 1F*). This conservation suggests that Ser[224] phosphorylation is potentially catalyzed by a conserved vertebrate kinase. Ergo, this PTM may contribute distinctly to the regulatory complexity of vertebrate genomes relative to other deuterostomes or protostomes.

## CTCF Ser[224] can be phosphorylated by CK2

Discovery of a novel phosphorylation site at Ser[224] raised the question as to which kinase could modify this position. It was previously reported that casein kinase 2 (CK2) could phosphorylate CTCF in vitro at Ser[604], Ser[609], Ser[610] and Ser[612] (*Klenova et al., 2001*). Interestingly, the highly conserved amino acid sequence surrounding Ser[224] fits the known CK2 recognition site (S-x-x-D/E) (*Figure 1F*) (*Meggio and Pinna, 2003*). Thus, we decided to test if CK2 also modifies Ser[224]. To do this, we first generated recombinant CTCF and in vitro phosphorylated it with CK2 and [$\gamma$-$^{32}$P]-ATP (*Sun et al., 2013*). Following SDS-PAGE, autoradiography revealed that recombinant CTCF can be phosphorylated by CK2 (*Figure 1—figure supplement 1A*). To determine if Ser[224] specifically is phosphorylated by CK2, we performed that assay with non-radioactive ATP and excised a Coomassie stained band corresponding to full length CTCF (red box, *Figure 1—figure supplement 1B*). Mass spectrometry analysis of the band, revealed that CKII phosphorylated CTCF Ser[224] (*Figure 1—figure supplement 1C*). As with our analysis of immunoprecipitated CTCF from MEFs, we did not observe phosphorylation at Ser[604], Ser[609], Ser[610] or Ser[612]. However, again, technical hurdles (i.e., the close proximity of these sites in a region poorly cut by trypsin and chymotrypsin) may preclude their identification by mass spectrometry. Furthermore, while we discovered that CK2 could phosphorylate Ser[224] in vitro, it remained to be seen if this occurs in vivo. Finally, it is also a distinct possibility that CTCF Ser[224] could be recognized by additional kinases pertinent to specific functions.

## Generation of a CTCF Ser[224]-P antibody

To explore the significance of this PTM in cells, we generated an antibody to CTCF Ser[224]-P in collaboration with Cell Signaling Technology. Following screening of prospective affinity purified antibodies, we further tested if the candidate CTCF Ser[224]-P antibody recognized a phosphorylated protein by western blot. We generated cell extract from immortalized rtTA MEFs and treated the extract with Lambda Protein Phosphatase. western blot revealed that the CTCF Ser[224]-P antibody had reduced affinity for the phosphatase treated samples (*Figure 2A*). The CTCF Ser[224]-P antibody also recognizes a smaller (~125 kDa) band that is not detected by a CTCF monoclonal antibody (*Figure 2A*). Notably CTCF migrates at a higher molecular weight (150 kDa) than predicted (84 kDa) in SDS-PAGE gels. One possibility is that CTCF Ser[224]-P occurs on two different CTCF isoforms in vivo, one of which is not recognized by the monoclonal antibody.

However, to further confirm that the affinity purified antibody preferred CTCF Ser[224]-P, we phosphorylated bacterially expressed recombinant CTCF and tested the antibody by western blot. We phosphorylated wild type CTCF, S224A, and S224E with CK2. While the western blot revealed some affinity of the CTCF Ser[224]-P antibody for unphosphorylated CTCF; the antibody preferred the CK2 phosphorylated protein (2[nd] column, *Figure 2B*). Moreover, the antibody is likely reactive to Ser[224]-P rather than other potential CK2 phosphorylation sites as there appeared to be no bias to CK2 phosphorylated S224A or S224E (*Figure 2B*).

## Cytological distribution of CTCF Ser[224]-P is comparable to total CTCF

Using immunofluorescence, we next asked if the CTCF Ser[224]-P antibody recognized its epitope in situ and if CTCF Ser[224]-P localization was comparable to total CTCF. Using immortalized tetraploid

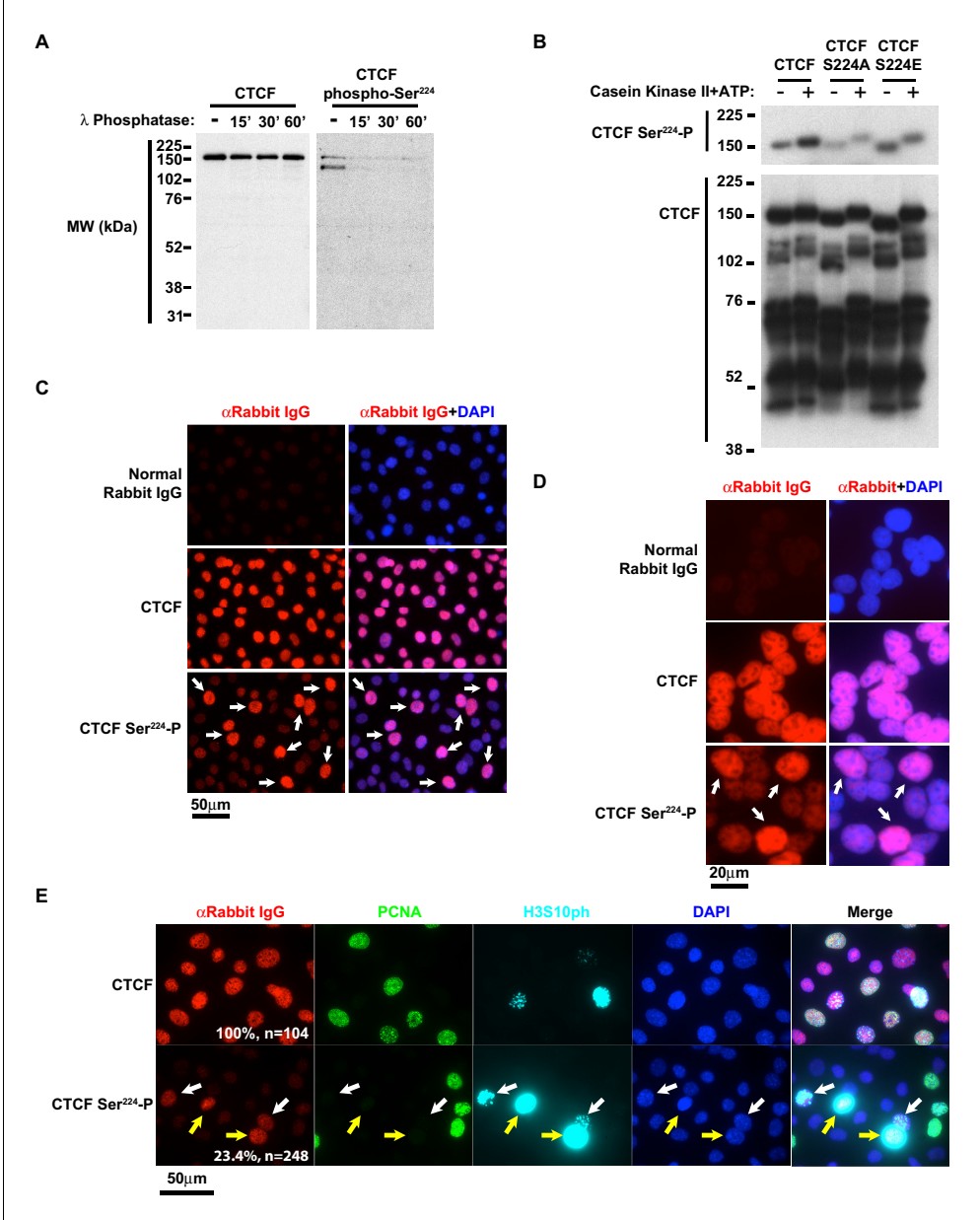

**Figure 2.** CTCF Ser224-P is conserved and accumulates at G2/M. (**A**) CTCF and CTCF Ser224-P western blots of MEF lysates treated with Lambda Protein Phosphatase for the indicated times. (**B**) CTCF and CTCF Ser224-P western blots of recombinant CTCF, CTCF S224A, and CTCF S224E ± Casein Kinase II in vitro phosphorylation. (**C**) Immunofluorescence performed on asynchronous MEFs with the indicated antibodies. Nuclei counterstained with DAPI. Bar, 50 μm. White arrows, prominent CTCF Ser224-P cells. (**D**) Immunofluorescence performed on asynchronous HEK293 with the indicated antibodies. Nuclei counterstained with DAPI. Bar, 20 μm. White arrows, prominent CTCF Ser224-P cells. (**E**) Immunofluorescence performed on asynchronous MEFs with the indicated antibodies. Nuclei counterstained with DAPI. Bar, 50 μm. % of n cells labeled with CTCF or CTCF Ser224-P antibodies indicated. White arrows, early G2. Yellow arrows, late G2.

DOI: https://doi.org/10.7554/eLife.42341.004

The following figure supplement is available for figure 2:

**Figure supplement 1.** CTCF Ser224-P accumulates in G2 and has a subnuclear distribution comparable to total CTCF.

DOI: https://doi.org/10.7554/eLife.42341.005

female MEFs, we found that the CTCF Ser$^{224}$-P antibody labeled a nuclear antigen in a pattern reminiscent of CTCF (*Figure 2C*, *Figure 2—figure supplement 1A*). Furthermore, CTCF was visually excluded during interphase from nucleoli and constitutive heterochromatic regions (*Burke et al., 2005*). Therefore we also scrutinized CTCF Ser$^{224}$-P relative to the nucleolus, constitutive heterochromatin, and facultative heterochromatin using antibodies against B23, H3K9me3 and H3K27me3 respectively (*Spector et al., 1984*; *Bannister and Kouzarides, 2011*). In wildtype female cells, H3K27me3 is known to be prominently enriched on the inactive X chromosome (Xi). In tetraploid MEFs, it usually appears as two nuclear bodies. CTCF Ser$^{224}$-P was not enriched on the Xi, and we found no evident visual distinction between CTCF Ser$^{224}$-P and total CTCF nuclear distribution regardless of co-staining nucleoli and heterochromatin (*Figure 2—figure supplement 1B–D*). Thus, at a cytological level, CTCF Ser$^{224}$-P appears to distribute similarly as total CTCF in the nucleus.

## CTCF Ser$^{224}$-P accumulates during the G2/M transition of the cell cycle

However, a remarkable observation from our immunofluorescence assays was that CTCF Ser$^{224}$-P labeling was not homogeneous in an asynchronous population of MEFs. Some of the nuclei were more intensely labeled by the CTCF Ser$^{224}$-P antibody whereas nuclei stained with total CTCF antibody were uniformly labeled (arrows, *Figure 2C*). While it also appears that there were less intensely CTCF Ser$^{224}$-P stained nuclei that were observable particularly when compared to normal IgG staining, this may be attributable to antibody cross reactivity to unmodified CTCF Ser$^{224}$ (*Figure 2B,C*). Since CTCF Ser$^{224}$-P potentially occurs in human cells (*Kettenbach et al., 2011*) and vertebrates share a high degree of sequence identity surrounding Ser$^{224}$ (*Figure 1E*), we next asked if the mottled CTCF Ser$^{224}$-P pattern is discernable in asynchronous human cells. As predicted, CTCF Ser$^{224}$-P labeling of asynchronous HEK293 revealed that a subset of cells was intensely labeled by the antibody (arrows, *Figure 2D*). This intimates that this PTM may be associated with a vertebrate-conserved CTCF regulatory mechanism.

Given the irregular CTCF Ser$^{224}$-P staining of asynchronous cells, we posited that perhaps this PTM is regulated by the cell cycle. To explore this possibility, we performed CTCF Ser$^{224}$-P immunofluorescence on asynchronous MEFs with additional antibodies to cell cycle markers. To identify cells in S phase and G2/M, we labeled cells with antibodies against PCNA and phosphorylated histone H3 Ser$^{10}$ (H3S10ph) respectively (*Bravo and Macdonald-Bravo, 1985*; *Hendzel et al., 1997*). While a total CTCF antibody labeled 100% of the cells (n = 104), again only a fraction of the cells (23.4%, n = 248) were CTCF Ser$^{224}$-P positive (*Figure 2E*). Comparing this fraction to the cell cycle markers revealed that a striking majority (94.8%) of the CTCF Ser$^{224}$-P positive cells were co-labeled with H3S10ph antibody (*Figure 2E*, *Figure 2—figure supplement 1E*). The CTCF Ser$^{224}$-P labeling was most prominent in cells in both early G2 and late G2/prophase, as evidenced from the H3S10ph staining pattern (white arrows, early G2; yellow arrows, late G2/prophase. *Figure 2E*, *Figure 2—figure supplement 1E*). There were also CTCF Ser$^{224}$-P positive cells co-stained with PCNA antibody (27.6%) (Orange and magenta arrows, *Figure 2—figure supplement 1E*). However, the majority of these CTCF Ser$^{224}$-P-PCNA double positive cells (81.3%) were also faintly positive for H3S10ph (magenta arrows, *Figure 2—figure supplement 1E*). Taken together, these findings indicate that phosphorylation of CTCF Ser$^{224}$ likely initiates at the end of S phase and peaks at G2/M. To further evaluate if CTCF Ser$^{224}$-P is enriched at G2/M, we arrested TST-1 mESCs 20 hours with the CDK1 inhibitor RO-3306 (*Vassilev et al., 2006*). RO-3306 treatment resulted in an accumulation of CTCF Ser$^{224}$-P positive cells blocked at G2/M as evident from the concomitant increase in H3S10ph co-staining (*Figure 2—figure supplement 1F*). These data pointed to an association of CTCF Ser$^{224}$-P with the G2/M transition of the cell cycle.

## PLK1 phosphorylates CTCF Ser$^{224}$

To explore how CTCF is specifically phosphorylated at the G2/M transition, we first sought to identify which kinase(s) could be phosphorylating CTCF Ser$^{224}$ at the end of S and in G2/M. Naturally, our first search criterion was that a prospective kinase also had to be expressed at these stages of the cell cycle. We also presumed that the kinase has a bona fide role limited to G2/M. Using these guidelines, Polo-like kinase I (PLK1) was the most attractive candidate. PLK1 is a regulator of the G2/M transition and phosphorylates mitosis-associated substrates (*Barr et al., 2004*). Importantly, PLK1 expression increases from S phase and peaks during G2 (*Golsteyn et al., 1994*; *Lake and Jelinek,*

_1993_). Comparing the consensus PLK1 substrate site to the amino acid sequence flanking CTCF Ser[224] revealed that Ser[224] is a potential target of PLK1 (_Figure 3A_) (_Nakajima et al., 2003_). Of note, CTCF was not identified as a PLK1 target in HeLa cells (_Kettenbach et al., 2011_). However, the designation of PLK1 targets in that study was not resultant from direct assay. Similarly, we did not find PLK1 in our CTCF-3xFLAG IP-MS/MS (_Figure 1D_) likely due to stringent nuclear extraction conditions used to extract CTCF from the chromatin fraction that disrupted native protein-protein interactions (see Materials and methods). We also examined PLK1 conservation across Nephrozoa by clustering the amino acid sequence identity of orthologs using Clustal. For reference, orthologs of the kinases ATM and GSK3B were also compared. We found that PLK1 conservation mirrors that of its potential CTCF Ser[224] substrate, where the kinase is most conserved amongst vertebrates (_Figures 1F_ and _3B_). This finding may further hint at a homologous CTCF Ser[224]-P function particular to vertebrates.

To test if CTCF can be phosphorylated by PLK1, we first treated asynchronous immortalized MEFs with the PLK1-specific inhibitor BI 6727 (_Rudolph et al., 2009_). We performed CTCF and CTCF Ser[224]-P immunofluorescence after 12 hr of treatment with 100 nM or 1000 nM BI 6727 ($EC_{50}$ ~10–150 nM) (_Rudolph et al., 2009_; _Rudolph et al., 2015_; _Gorlick et al., 2014_). Consistent with inhibition of PLK1, we observed a likely G2/M defect evident in a higher percent of H3S10ph positive cells in BI 6727 treated cells than control (DMSO, 5.6%; 100 nM BI 6727, 11.4%; 1000 nM BI 6727, 29.2%; t = 12 hr; n > 100 cells). While 30.8% (n = 214) of untreated cells were CTCF Ser[224]-P positive, only 18.3% (n = 175) were positive following treatment with 100 nM BI 6727 (_Figure 3C_). The range of $EC_{50}$ values for BI 6727 (10–150 nM) suggests that remaining CTCF Ser[224]-P was likely due to incomplete inhibition of PLK1 at this concentration. Supporting this, 1000 nM BI 6727 completely ablated CTCF Ser[224]-P phosphorylation (n = 101) (_Figure 3C_). Total CTCF expression was not affected by 12 hr treatment with BI 6726 at either concentration (_Figure 3C_). While this indicates that PLK1 is essential for CTCF Ser[224]-P, treatment with BI 6727 does not exclude the possibility that our observations were an indirect outcome. Therefore, we also performed an in vitro kinase assay with purified recombinant CTCF and PLK1, using both Casein and PLK1 autophosphorylation as positive controls (*** and ** respectively, _Figure 3D_) (_Golsteyn et al., 1995_). In support of our hypothesis, we found that PLK1 directly phosphorylated CTCF (*, _Figure 3D_). However, this result did not indicate if CTCF Ser[224]-P had occurred. To test this, we performed the PLK1 in vitro kinase assay without radioactive isotope and we excised the band corresponding to full-length CTCF from the Coomassie stained gel (red box, _Figure 3E_). Analysis by mass spectrometry revealed phosphorylation at Ser[224], arguing that indeed PLK1 is the kinase for CTCF Ser[224] (_Figure 3F_). However, as we did not capture in vivo interactions of CTCF with PLK1 (or CK2), we could not confirm either as the CTCF kinase for certain. Therefore, both PLK and CK2 remain potential CTCF kinases.

## CTCF Ser[224]-P localizes to pericentric regions during mitosis

We posited that this modification could be observed on condensed chromatids during metaphase. It was previously reported that CTCF remains bound to mitotic chromosomes – both along the dyad arms and at centromeres (_Burke et al., 2005_; _Rubio et al., 2008_). However it was also reported that CTCF is phosphorylated in its zinc finger domain during mitosis which results in its dissociation from chromatids (_Sekiya et al., 2017_). Notably, the latter study did not directly identify phosphorylation sites or visualize the mitotic chromatids. Accordingly, to resolve this discord and test our hypothesis, we performed immunofluorescence on mESC metaphase spreads alongside antibody to the facultative heterochromatin mark H3K27me3 that is found on chromatid arms but is excluded from the centromere (_Terrenoire et al., 2010_). Much like CTCF, CTCF Ser[224]-P could be observed on chromatid arms, albeit not uniformly like CTCF (_Figure 3G_, _Figure 3—figure supplement 1A–B_). Remarkably, CTCF Ser[224]-P appeared distinctly enriched in between the centromeres of the murine acrocentric chromosomes and the H3K27me3 positive dyad arms. We therefore concluded that the metaphase localization of CTCF Ser[224]-P is pericentric (_Figure 3G–H_, _Figure 3—figure supplement 1A–B_). As PLK1 is associated with kinetochores, it is possible that PLK1 phosphorylates CTCF in the vicinity of the centromere for a mitosis function (_Barr et al., 2004_).

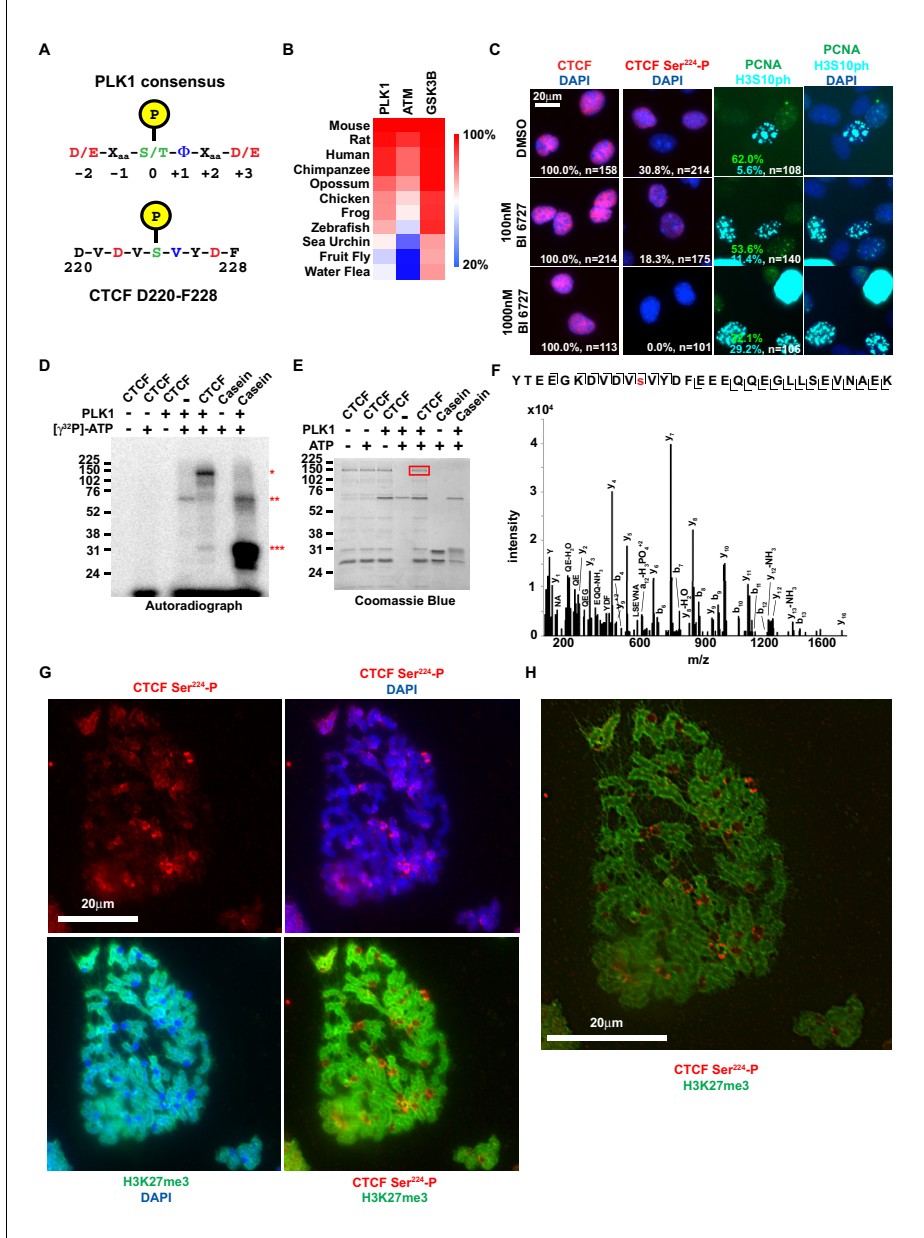

**Figure 3.** CTCF Ser[224] is phosphorylated by PLK1 and prominently labels pericentric chromatin. (**A**) Graphic comparison of PLK1 consensus substrate sequence with CTCF D220-F228. Green S/T with yellow encircled P, phosphorylation site at position 0. Red D/E, aspartic or glutamic acid. Blue Φ, hydrophobic amino acid. (**B**) Amino acid sequence identity heat map for the conserved kinases PLK1, ATM, and GSK3B. Nephrozoa species aligned in *Figure 1F* shown. (**C**) CTCF, CTCF Ser[224]-P, PCNA, and H3S10ph immunofluorescence performed on MEFs treated with DMSO or BI 6727 at the indicated concentrations for 12 hr. Nuclei counterstained with DAPI. Bar, 20 μm. % of n cells labeled with CTCF, CTCF Ser[224]-P, PCNA, or H3S10ph antibodies indicated. (**D**) PLK1 in vitro kinase assay with CTCF or dephosphorylated Casein substrates. Red *, phosphorylated CTCF. Red **, autophosphorylated PLK1. Red ***, phosphorylated casein. (**E**) In vitro kinase assay performed in parallel to (**D** without radioactive isotope. SDS-PAGE gel Coomassie stained. Red box, CTCF band excised for mass spectrometry analysis. (**F**) Manually validated mass spectra of CTCF peptide Tyr[214]-Lys[244] with y and b ions identified. Phosphorylation event at Ser[224] indicated by red s. (**G**) Immunofluorescence performed on TST-1 mESC metaphase chromosomes with the indicated antibodies. DNA stained with DAPI. (**H**) CTCF Ser[224]-P and H3K27me3 co-stain from (**G**) deconvolved.
DOI: https://doi.org/10.7554/eLife.42341.006

The following figure supplement is available for figure 3:

*Figure 3 continued on next page*

*Figure 3 continued*

**Figure supplement 1.** CTCF Ser[224]-P is enriched at pericentric regions of metaphase chromosomes.
DOI: https://doi.org/10.7554/eLife.42341.007

## CTCF Ser224-P binds to a subset of CTCF binding sites during interphase

As the phospho-specific antibody labeled metaphase chromatin and thus demonstrated in vivo DNA association of CTCF Ser[224]-P, we next determined its precise genome-wide distribution. To this end, we performed both CTCF Ser[224]-P and CTCF chromatin immunoprecipitation and sequencing (ChIP-seq) on asynchronous TST-1 mESCs. From our CTCF ChIP-seq, we detected ~50,000 CTCF peaks genome-wide (z = 6). MEME-ChIP analysis tellingly revealed significant enrichment of CTCF motifs (JASPAR MA0139.1) centered in these peaks (54% of peaks, p=3.0e-10948) (*Figure 4—figure supplement 1A*) (*Bailey et al., 2009*). In contrast, only ~900 CTCF Ser[224]-P peaks were detected in our CTCF Ser[224]-P ChIP-seq (z = 6). Using MEME-ChIP, we de novo identified a significantly enriched motif in the CTCF Ser[224]-P peaks (48% of peaks, p=2.7e-88) that closely matched the CTCF motif (p=1.8e-86) and was likewise centered in the peaks (*Figure 4A*, *Figure 4—figure supplement 1B*).

Further CEAS analysis of the CTCF Ser[224]-P peaks revealed a genomic feature distribution that was also broadly similar to that of CTCF peaks (*Figure 4B*) (*Shin et al., 2009*). The distributions of CTCF and CTCF Ser[224]-P peaks were also spread across all chromosomes (*Figure 4—figure supplement 1C,D*). Notably, the vast majority (95.9%) of the CTCF Ser[224]-P peaks intersected with our CTCF peaks (*Figure 4C*). And comparison to mESC RNA-seq data (GSM723776) revealed that CTCF Ser[224]-P peaks were also proximal to both transcribed and non-transcribed regions (*Figure 4C*) (*Shen et al., 2012*). Binding to pericentric sequences was difficult to detect in deep-sequencing based assays because of their highly repetitive nature. Knowing that our CTCF Ser[224]-P antibody showed a low level of cross-reactivity to unmodified CTCF (*Figure 2A,B*), we next wondered whether our CTCF Ser[224]-P binding could be explained by this cross-reactivity. Accordingly, we examined the correlation between CTCF Ser[224]-P and CTCF binding as detected by our ChIP-seq (*Figure 4D*). While there was a detectable linear relationship between CTCF Ser[224]-P and CTCF binding signal, this was only sufficient to explain about half of all CTCF Ser[224]-P binding ($R^2$, fraction of variation in CTCF Ser[224]-P binding explained by CTCF binding, for all peaks = 0.48, $R^2$ for shared peaks = 0.54). Therefore, at least a subset of CTCF Ser[224]-P ChIP-seq peaks likely represented real binding sites outside of pericentric regions. In addition, as our ChIP-seq was performed in unsynchronized ES cells, which are mostly in interphase (S phase), this suggests that CTCF Ser[224]-P is bound at these regions outside of G2/M as well.

We next sought to examine features which differentiated regions bound by CTCF Ser[224]-P from regions bound by unmodified CTCF. Shared CTCF and CTCF Ser[224]-P ChIP peaks (representing 95.9% of all CTCF Ser[224]-P peaks) had significantly more CTCF binding signal than CTCF peaks in general (*Figure 4E*, p<2.2e-16, Wilcoxon rank sum test). While CTCF motifs found in CTCF Ser[224]-P peaks did not tend to be more conserved than motifs found in CTCF ChIP peaks (*Figure 4—figure supplement 1E*, p=0.1623, Wilcoxon rank sum test), CTCFSer[224]-P ChIP peaks did tend to contain more motifs than CTCF ChIP peaks (*Figure 4F*, p<2.2e-16, Wilcoxon rank sum test). In other words, CTCFSer[224]-P ChIP peaks tend to be found at higher affinity sites with greater numbers of CTCF binding sites than CTCF ChIP peaks in general. We also found that shared CTCF and CTCF Ser[224]-P ChIP peaks tended to be larger than CTCF peaks in general (*Figure 4—figure supplement 1F*, p<2.2e-16, Wilcoxon rank sum test), which may in part explain these trends.

Finally, we compared our CTCF and CTCFSer[224]-P ChIP-seq with a previously published cohesin ChIP-seq done in mESCs (*Kagey et al., 2010*). While 22.8% and 33.3% of our CTCF ChIP-seq peaks overlapped with SMC1 and SMC3 peaks, respectively, 51.6% and 85.1% of our CTCFSer[224]-P ChIP peaks overlapped with SMC1 and SMC3 peaks, respectively. This suggests that CTCFSer[224]-P ChIP tends to co-bind more frequently with cohesin as well.

## Mutational analysis of CTCF Ser[224]-P reveals an effect on cell growth

We next decided to examine the effect of expressing either Ser[224] mutants, S224A or S224E, on cells. The former mutation obviates phosphorylation and the latter glutamic acid substitution is a so-

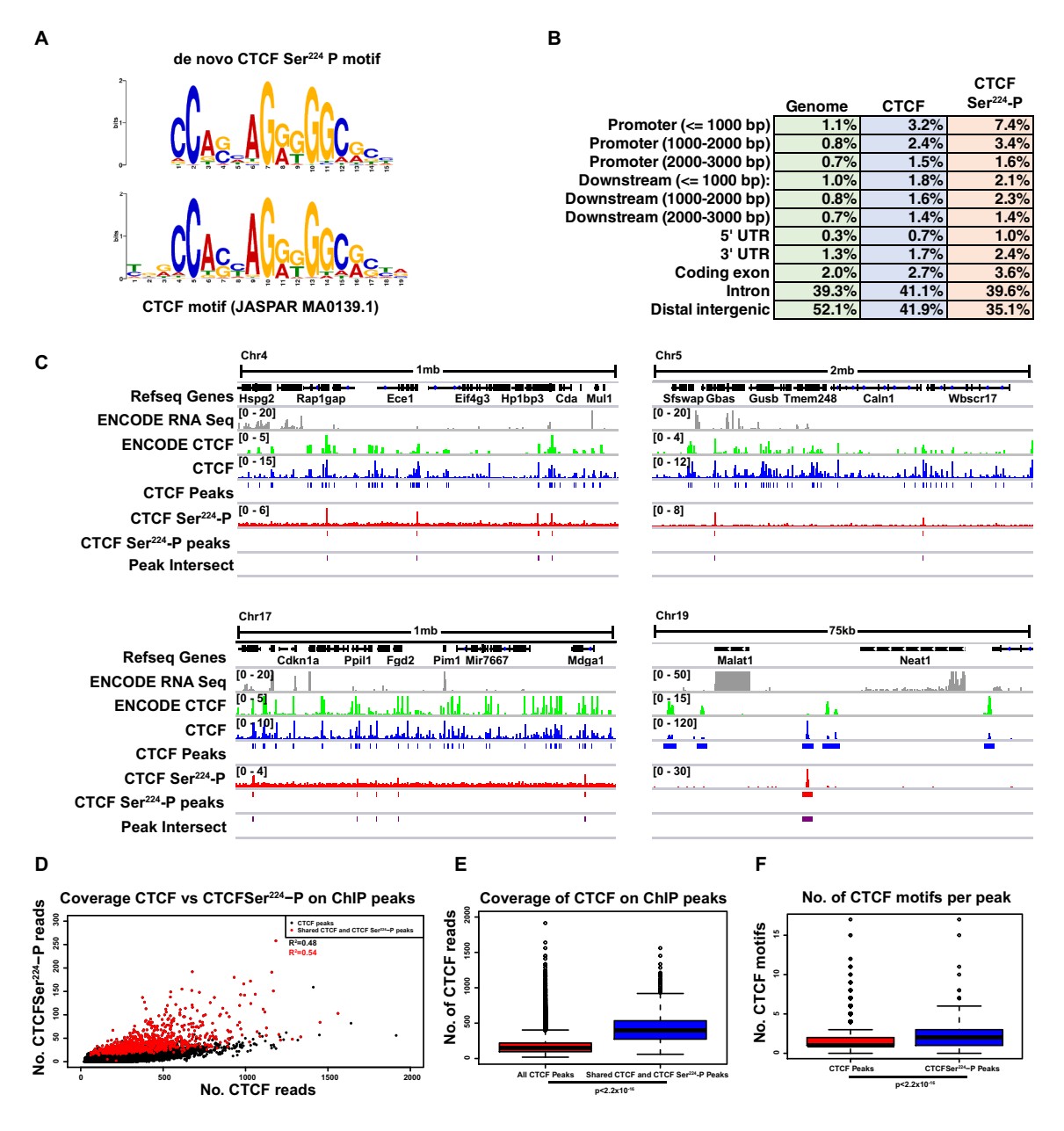

**Figure 4.** CTCF Ser[224]-P occupies a fraction of CTCF sites outside of pericentric chromatin in interphase. (**A**) De novo CTCF Ser[224]-P motif logo determined with MEME-ChIP (top). JASPAR indexed CTCF motif logo (bottom). (**B**) % distribution of CTCF (blue) and CTCF Ser[224]-P (orange) ChIP-seq peaks across genomic features. % distribution of these features in the genome is shown for comparison (green). (**C**) Four example screenshots showing CTCF (blue) and CTCF Ser[224]-P (red) ChIP-seq coverage tracks and called peaks. Intersected CTCF and CTCF Ser[224]-P peaks (purple) are also shown. ENCODE CTCF ChIP-seq coverage (green), Refseq Genes (black) and ENCODE RNA-seq (gray) are shown for reference. Chromosome number and window scale are indicated. (**D**) CTCF versus CTCF Ser[224]-P ChIP-seq coverage on CTCF (black) and shared CTCF and CTCF Ser[224]-P ChIP-seq peaks (red). R-squared values for both sets of peaks are shown. (**E**) CTCF ChIP-seq coverage on CTCF versus shared CTCF and CTCF Ser[224]-P ChIP-seq peaks ($p < 2.2 \times 10^{-16}$, Wilcoxon rank sum test). (**F**) Number of CTCF motifs found in CTCF versus shared CTCF and CTCF Ser[224]-P ChIP-seq peaks ($p < 2.2 \times 10^{-16}$, Wilcoxon rank sum test).

DOI: https://doi.org/10.7554/eLife.42341.008

The following figure supplement is available for figure 4:

**Figure supplement 1.** CTCF Ser[224]-P occupies known CTCF binding sites.
DOI: https://doi.org/10.7554/eLife.42341.009

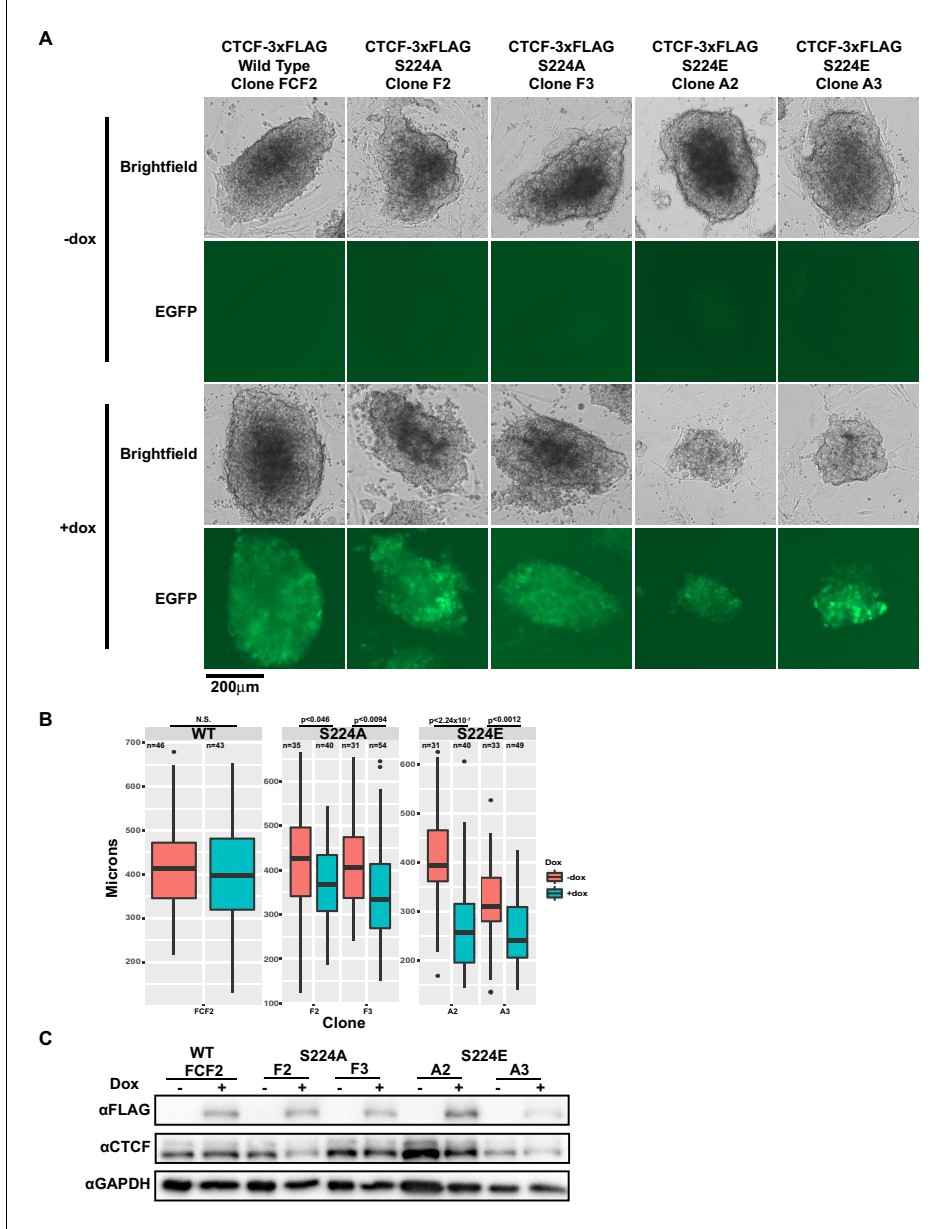

**Figure 5.** CTCF S224E phosphomimic mutation is poorly tolerated by dividing cells. (**A**) Representative brightfield and EGFP images of F1-2.1 mESCs carrying a dox-inducible wild type CTCF, S224A or S224E transgene grown for six days with (bottom) or without (top) doxycycline. Two independent S224A and S224E clones are shown. (**B**) Quantification of colony diameters in microns of cell lines shown in (**A**), grown for six days with (blue) or without (red) doxycycline. Student's t-test was used to calculate p-values between indicated samples, with not significant (N.S.) p-values being >0.05. (**C**) Western blot measuring FLAG and CTCF protein levels of cell lines shown in (**A**). GAPDH is shown as a loading control.

DOI: https://doi.org/10.7554/eLife.42341.010

called phosphomimetic that mimics negatively charged phosphorylation by presenting an acidic side chain at that position. We generated doxycycline-inducible S224A- and S224E-3xFLAG mESCs and overexpressed wild type CTCF, S224A or S224E over six days of growth. Ectopic expression of S224E but not wild type or S224A affected growth of mESCs (**Figure 5A**). Namely, mESC colonies overexpressing S224E for 6 days had significantly smaller diameters than uninduced cells, while over-expressing wild type or S224A did not result in significantly smaller colonies (**Figure 5B**). Expression

levels of the induced FLAG-tagged proteins were comparable in all clones by western blot (*Figure 5C*). Taken together, our data so far suggest that the phosphorylated form of CTCF may have a specific function during the cell cycle, as forced constitutive expression of the phosphomimetic form results in a cell growth defect. However, we also note that the continued presence of endogenous wild-type CTCF in our overexpression system may be obscuring detection of further phenotypes of S224A and S224E CTCF.

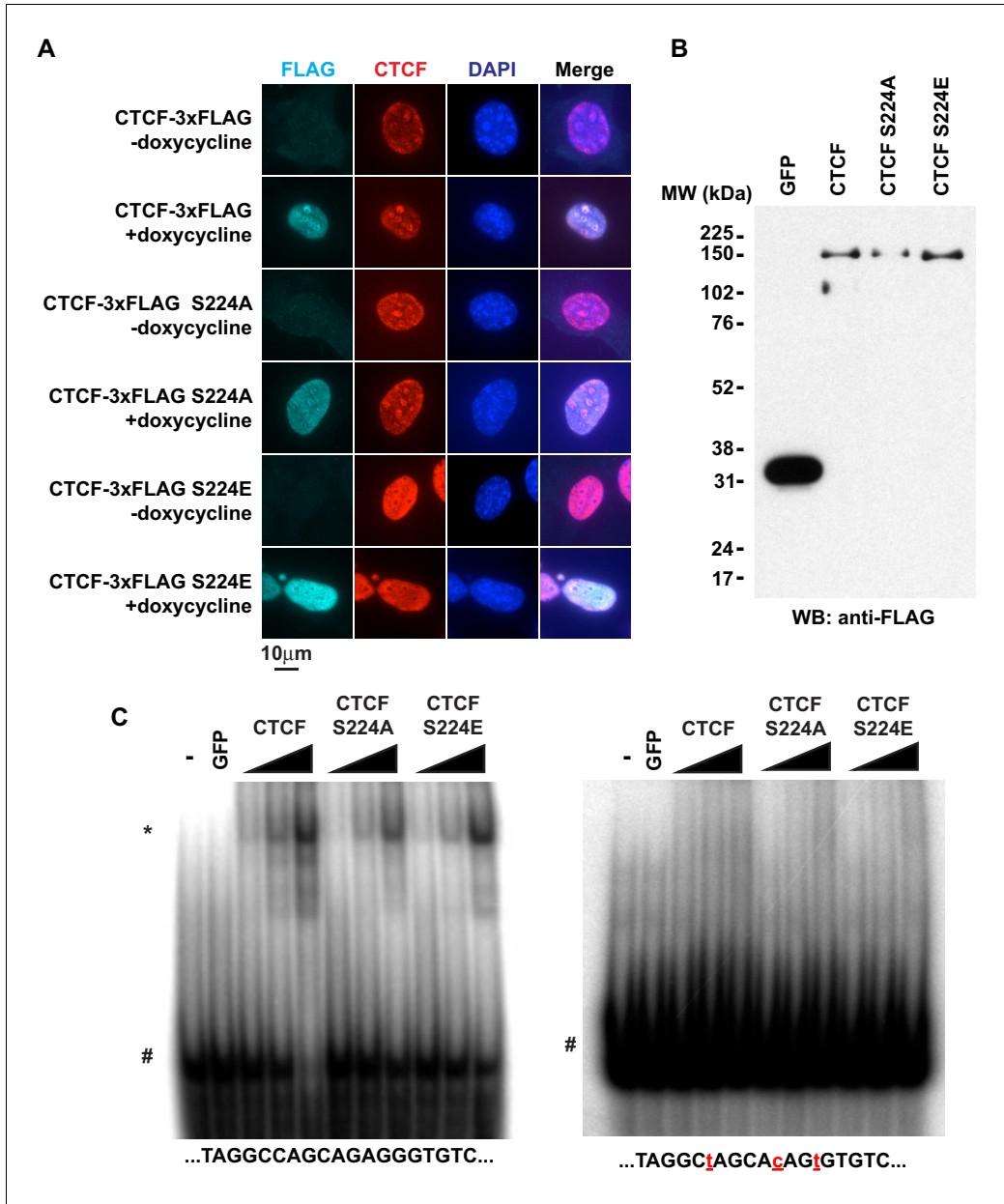

**Figure 6.** CTCF Ser$^{224}$ is nonessential to nuclear import and DNA binding. (**A**) FLAG and CTCF immunofluorescence performed on rtTA MEFs with inducible CTCF-3xFLAG transgenes (wild type, S224A, or S224E). Nuclei counterstained with DAPI. Bar, 10 µm. (**B**) FLAG western blot of recombinant FLAG-tagged GFP and CTCF (wild type, S224A, or S224E). (**C**) DNA EMSA using RS14C (left) and RS14C mutant (right) probes. Red lowercase letters, mutated positions. 2 pmole GFP and 0.5, 1, or 2 pmole CTCF (wild type, S224A, or S224E) were used. *, CTCF shifted probe. #, free probe.
DOI: https://doi.org/10.7554/eLife.42341.011

## CTCF Ser[224] mutations have no effect on nuclear import, DNA binding, cell cycle, or ploidy

Given the phenotype, we further explored S224 mutants to understand the normal function of CTCF Ser[224]-P. We initially posited that CTCF Ser[224]-P may be critical to regulating some chromatin function of CTCF. As CTCF Ser[224]-P was observable in the nucleus in only a fraction of asynchronous cells, it was a remote possibility that Ser[224] is also critical for CTCF nuclear import. Thus, we examined localization of inducible wild type CTCF, S224A, and S224E in MEFs by immunofluorescence. Regardless of either amino acid substitution at Ser[224], nuclear localization of CTCF was not affected. (*Figure 6A*). As we observed CTCF Ser[224]-P bound to only a fraction of CTCF sites genome-wide, it was also a possibility that this amino acid position is critical for DNA binding. However, since Ser[224] is not located in the zinc finger domain, we predicted that Ser[224] would not directly influence sequence-specific DNA binding. To test this, we performed EMSA using a dsDNA probe with a known CTCF motif as well as a probe with mutations in the motif (*Spencer et al., 2011*). We made recombinant FLAG-tagged S224A and S224E as well as wild type CTCF and GFP as positive and negative controls respectively (*Figure 6B*). As expected, regardless of amino acid polarity or charge, both mutants bound the known CTCF motif and not the mutated motif (*Figure 6C*). However, it is still a distinct possibility that, in vivo, CTCF Ser[224]-P may signify a regulatory event that determines which of the thousands of genomic CTCF sites are bound.

To test whether CTCF Ser[224]-P may play a role on mitotic chromosomes, we asked if could overexpression of S224A or S224E leads to defects in either cell cycle progression or segregation of chromatids. We generated DNA content profiles of mESCs with inducible wild-type CTCF, S224A, or S224E transgenes after six days of doxycycline induction and did not observe any blockages in the cell cycle (*Figure 7A*). We also investigated changes in ploidy by kayotyping these cells, specifically examining metaphase spreads for polyploidy and translocations. However, we did not find significantly higher numbers of polyploid spreads in mESCs overexpressing wild-type or mutant CTCF versus uninduced mESCs (*Figure 7B,C*).

## CTCF Ser[224]-P and the borders of topologically associating domains (TADs)

Finding that overexpression of wild type, S224A or S224E had little obvious impact on mitotic chromosomes, we investigated whether it could interfere with CTCF function in interphase. We first noted that many of our CTCF Ser[224]-P ChIP-seq peaks detected in interphase overlapped CTCF peaks at the borders of Topologically Associating Domains (TADs) (*Figure 8A*), megabase-scale organizational structures on chromosomes within which genetic elements show high frequency of interaction (*Dixon et al., 2012*; *Nora et al., 2012*). Mammalian chromosomes are generally organized into hundreds of such TADs, with each TAD separated by genetically defined 'borders'. As previous studies had shown that CTCF binding is important for formation of TAD borders (*Sanborn et al., 2015*; *Nora et al., 2017*), we decided to examine the impact of overexpressing wild-type CTCF, S224A or S224E on nuclear architecture using HYbrid Capture Hi-C (Hi-C[2]), a cost-effective alternative to genome-wide Hi-C (*Sanborn et al., 2015*). The TAD containing the gene *Mecp2* was chosen as the capture region as it contained a sub-TAD domain bound by CTCF at both the left and right borders and CTCF Ser[224]-P at the left border (*Figure 8A*). To minimize secondary impacts on TAD structure, Hi-C was performed after 2 days of wild type CTCF, S224A or S224E overexpression in F1-2.1 mESCs, a time point at which no cell colony defects were observed in any of the three cell lines. By eye, interaction matrices of the *Mecp2* region appeared similar with or without overexpression of wild type CTCF, S224A and S224E (*Figure 8A*). To analyze impact on the interactions within the *Mecp2* TAD quantitatively, we additionally calculated insulation scores across the Hi-C[2] region and used this to calculate a TAD score for the *Mecp2* TAD in each condition (*Crane et al., 2015*). The *Mecp2* TAD score was similar with or without overexpression of wild type CTCF, S224A and S224E. Thus, overexpression of CTCF, including CTCF S224E, does not detectably impact three-dimensional chromatin structure (*Figure 8B*).

However, as RNA-seq suggested that levels of S224E CTCF may be modest as compared to wild-type CTCF (see below), we assayed to what extent the mutant S224E CTCF was bound to the Mecp2 TAD. We did CTCF and FLAG ChIP-qPCR in doxycycline inducible S224E-3xFLAG mESCs after 48 hr of dox induction, the same conditions under which the Hi-C[2] experiment was performed

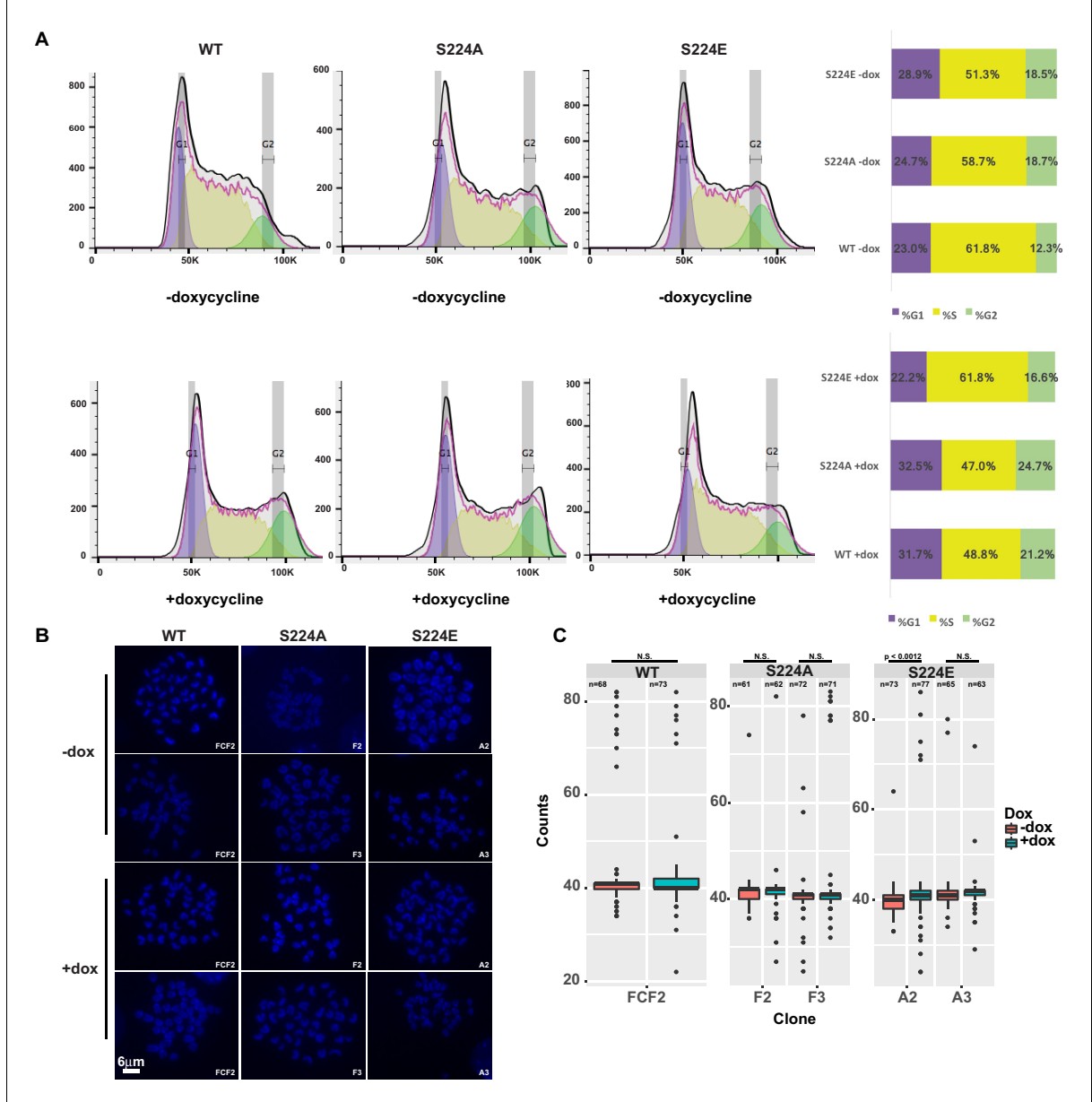

**Figure 7.** Overexpression of CTCF, S224A or S224E does not impact cell cycle progression or ploidy. (**A**) Cell cycle profiles of F1-2.1 mESCs carrying a dox inducible CTCF, S224A or S224E transgene grown for six days with (bottom) or without (top) dox induction. Quantification of percent of cells in each stage of the cell cycle indicated in bar graph to the right. (**B**) Representative metaphase spreads of cells profiled in *A* grown for six days with (bottom) or without (top) dox induction. (**C**) Quantification of chromosome counts for cells profiled in *A* grown for six days with (bottom) or without (top) dox induction. Wilcoxon rank sum test was used to calculate p-values between indicated samples, with not significant (N.S.) p-values being >0.05.
DOI: https://doi.org/10.7554/eLife.42341.012

(*Figure 8C*). While FLAG binding was enriched at positive control region Cirbp over IgG, the –dox sample and a negative control region Oct4 (*Figure 8C*), the level of binding was much lower than that of CTCF. Similarly, FLAG binding was modestly enriched over IgG, the –dox sample, and Oct4 at at least the two CTCF sites bordering the Mecp2 TAD (*Figure 8C*, Irak1, Ikbkg), as well as at the site bound by phospho-CTCF in the Mecp2 TAD in one replicate (*Figure 8C*, Flna), although to an extent much less than that of CTCF. This modest binding of S224E CTCF to the Mecp2 TAD, likely due to endogenous wild-type CTCF remaining in the system, may possibly explain why we were unable to detect changes in chromatin architecture in this region.

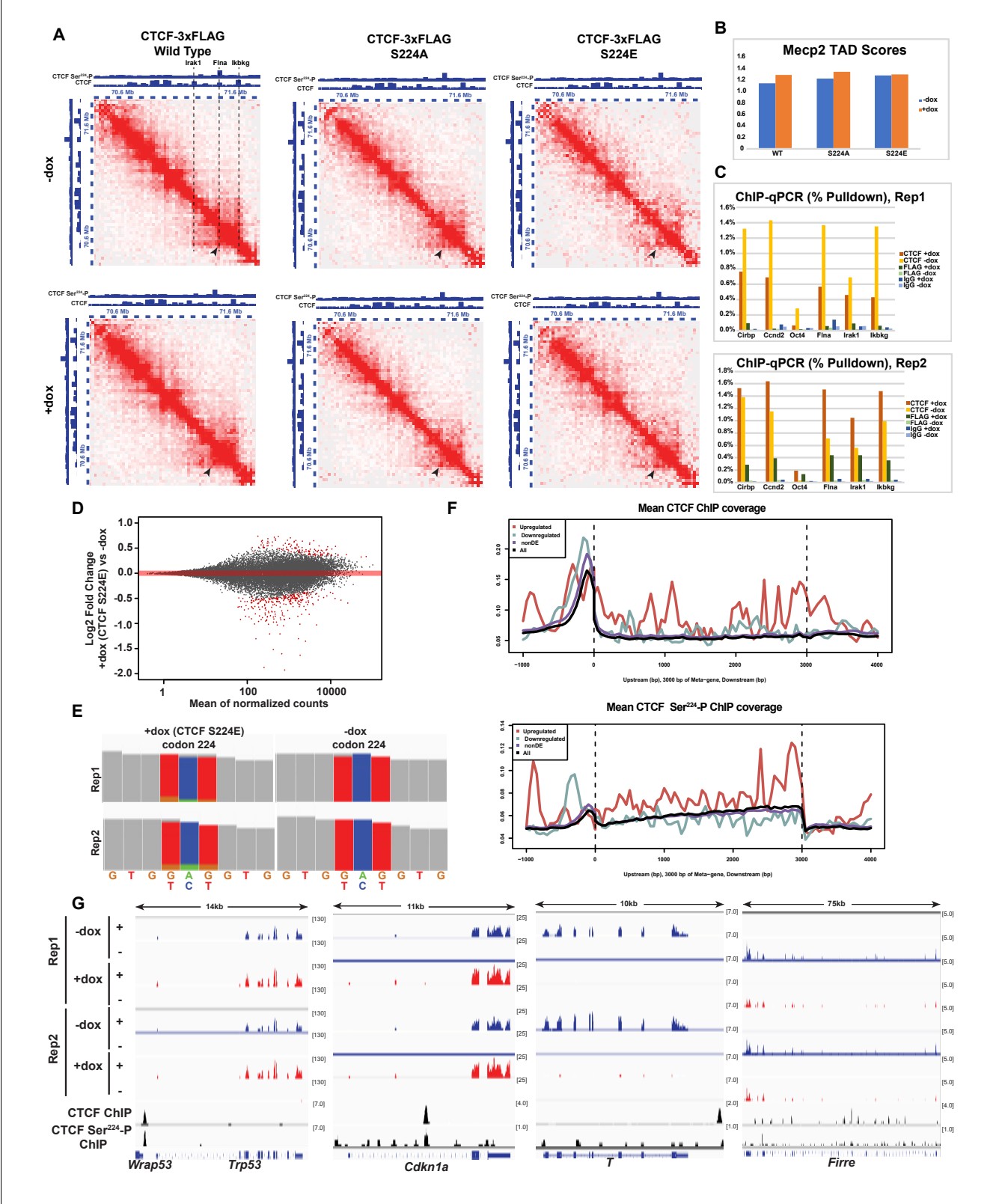

**Figure 8.** Impact of overexpression of CTCF, S224A and S224E on three-dimensional chromatin structure and gene expression. (**A**) Hi-C$^2$ interaction maps at 25 kb resolution of the Mecp2 TAD in F1-2.1 mESCs carrying dox-inducible wild type, S224A or S224E CTCF-3xFLAG transgenes grown for 2 days with (bottom) and without (top) doxycycline. CTCF and CTCF Ser$^{224}$-P ChIP-seq tracks are shown for comparison. Black arrows indicate the left border of a sub-TAD domain bound at both borders by CTCF and at one border by CTCF Ser$^{224}$-P. In addition, dotted lines and text in the WT -dox

*Figure 8 continued on next page*

*Figure 8 continued*

Hi-C$^2$ interaction map indicate locations of ChIP-qPCR primers used in (**C**), with the Irak1 and Ikbkg lines also indicating the borders of the Mecp2 TAD scored in (**B**). (**B**) TAD scores for the Mecp2 TAD for the Hi-C$^2$ interaction maps in *A*, with higher TAD score indicating a stronger TAD. (**C**) CTCF and FLAG ChIP-qPCR (% Pulldown) in F1-2.1 mESCs carrying dox-inducible S224E CTCF-3xFLAG grown for 2 days with or without doxycycline. Cirbp and Ccnd indicate positive control regions for CTCF binding, while Oct4 indicates a negative control region. Flna, Irak1, and Ikbkg regions are as indicated in (**A**). (**D**) MA plot of RNA-seq expression changes in F1-2.1 mESCs carrying dox-inducible CTCF S224E transgene after 6 days of dox induction. Points in red are DE genes (adjusted p-value<0.01). (**E**) Coverage of RNA-seq reads over codon 224 of CTCF with (left) and without (right) 6 days of dox induction. The number of reads with A (green), C (blue), G (gold), T (red) or N (grey) at positions 1, 2, and 3 of codon 224 are shown. (**F**) Metagene coverage of CTCF ChIP-seq reads (top) and CTCF Ser$^{224}$-P ChIP-seq reads (bottom) over upregulated (red), downregulated (green), non differentially expressed (purple) and all (black) genes. (**G**) RNA-seq, CTCF and CTCF Ser$^{224}$-P ChIP-seq coverage over two representative upregulated (left) and downregulated (right) genes.

DOI: https://doi.org/10.7554/eLife.42341.013

## Overexpression of CTCF S224E upregulates the p53 signaling pathway and globally plasma membrane proteins at the RNA level

As we could not attribute the mESC growth phenotype to changes in cell cycle progression, CTCF binding or TAD structure, we performed RNA sequencing (RNA-seq) to identify genes that were differentially expressed (DE) when CTCF S224E was induced in mESC's. We found 375 genes that were DE between two replicates (adjusted p-value<0.01), with 118 (31%) being upregulated and 257 (69%) being downregulated upon overexpression of CTCF S224E (*Figure 8C*). Surprisingly, CTCF was not a DE gene. To confirm that CTCF S224E was expressed, we examined RNA-seq reads overlapping CTCF codon 224 and found that 12–16% of reads had the glutamate codon at position 224 (*Figure 8D*). This indicates that while CTCF S224E was present, it was not grossly overexpressed at the RNA level; therefore, our observations from modest exogenous expression may reflect the importance of this site. We next looked at coverage of CTCF and CTCF Ser$^{224}$-P ChIP-seq over DE genes in an attempt to determine whether DE genes were directly regulated by CTCF. On average, CTCF binding was enriched along both downregulated and upregulated DE genes, albeit at different locations. In particular, downregulated genes were more enriched in CTCF binding upstream of the transcription start site, while upregulated genes were more enriched in CTCF binding along the gene body. CTCF Ser$^{224}$-P enrichment followed a similar pattern (*Figure 8E*). Next, in order to determine if specific pathways were enriched among our DE genes, we performed functional annotation analysis of upregulated and downregulated DE genes using the PANTHER Overrepresentation Test (*Mi et al., 2017*; *Nikolsky and Bryant, 2009*), with the list of all expressed genes as the background. Among upregulated genes, the only significantly overrepresented PANTHER pathway was the p53 pathway (FDR = 0.0495), most notably including *Trp53* (upregulated 1.3-fold, adjusted p-value<0.00075) and *Cdkn1a* (upregulated 1.3-fold, adjusted p-value<0.00043), genes which were also bound by CTCF and CTCF Ser$^{224}$-P (*Figure 8F*). Meanwhile, among downregulated genes, the most significant overrepresented cellular component categories were 'cell periphery' (FDR = $2.28 \times 10^{-6}$) and 'plasma membrane' (FDR = $3.45 \times 10^{-6}$), although, many downregulated genes did not fall into these categories (*Figure 8F*). These results suggest that modest CTCF S224E overexpression could lead to the slight activation of p53 signaling, and also global downregulation of proteins on the plasma membrane. The effect may be direct, as CTCF binding was found to be enriched at DE genes or indirect, as RNA-seq was assayed on cells after six days of S224E overexpression in mESCs, when a colony grown phenotype was already apparent. We conclude that the serine 224 of CTCF plays a critical role in regulating gene expression and suggest that the effect on the p53 signaling pathway may partly explain the growth phenotype in mESCs.

## Discussion

In the quarter century since its discovery, CTCF has been ascribed a multitude of chromatin functions. While DNA sequence composition and CpG methylation are determinants of CTCF binding, a regulatory system independent of DNA binding has remained elusive. Here we discovered and confirmed a CTCF PTM conserved among vertebrates. We observe for the first time that CTCF is differentially regulated during the cell cycle, with CTCF Ser$^{224}$-P being enriched at G2/M and specifically on pericentric regions during metaphase. Notably, CTCF Ser$^{224}$ phosphorylation is regulated by the

kinase PLK1 whose expression profile also peaks in G2/M. As exogenously expressing S224E leads to colony formation defects and gene expression changes, our data hint at the importance of CTCF Ser$^{224}$-P for proper cell function. We were further able exclude several potential functions of CTCF Ser$^{224}$-P. First, Ser$^{224}$ does not directly affect CTCF binding to DNA. However, while CTCF Ser$^{224}$-P occupies a subset of all CTCF sites genome wide, how this PTM is limited to a fraction of sites on the chromatids and pericentric satellites is still unclear. Second, while overexpression of a S224E leads to colony formation defects, we did not observe any changes in a TAD's structure, suggesting that CTCF Ser$^{224}$-P may not impact the architectural role of CTCF, at least around the gene-rich *Mecp2* TAD. Third, cell cycle progression and ploidy of cells overexpressing the phospho-mimic were not obviously affected. We conclude that regulation of CTCF Ser$^{224}$-P is needed to prevent dysregulation of hundreds of genes, directly or indirectly through mechanisms that likely do not involve CTCF interphase binding, architectural function, cell cycle progression or ploidy. However, an intriguing possibility which we did not investigate is that the negative charge introduced by phosphorylation at Ser224 could alter the conformation of the CTCF protein, altering the manner in which CTCF regulates genes or interacts with other proteins. We hope these observations will provide starting points for further elucidating the natural functions of this CTCF PTM and valuable resources for future study.

With our observation of CTCF Ser$^{224}$-P enrichment proximal to the centromere, several avenues of investigation become conspicuous. First, what could be the pericentric function of CTCF Ser$^{224}$-P? For example, CTCF interacts with cohesins and the latter remain at the centromere to facilitate cohesion during metaphase (*Rubio et al., 2008*; *Xiao et al., 2011*; *Stedman et al., 2008*; *Morales and Losada, 2018*). CTCF has also been found to impede 'loop extrusion' activity of the cohesin complex during interphase (*Vian et al., 2018*). Could a similar capacity be expected for pericentric CTCF Ser$^{224}$-P and the cohesin complex? Likewise, the presence of pericentric CTCF Ser$^{224}$-P may demarcate a topological boundary between the chromatid arms and satellite DNA. Notably, PLK1 also phosphorylates cohesins, promoting their prophase dissociation from the chromatid arms (*Losada et al., 2002*; *Hauf et al., 2005*; *Giménez-Abián et al., 2004*; *Sumara et al., 2002*). PLK1 phosphorylation of CTCF may also be linked to cohesin dissociation from the chromatids. Another prospective undertaking would be to determine if CTCF Ser$^{224}$-P is present at all pericentromeres. Are all pericentromeres identical? Thus, the identification of this PTM and development of a specific antibody may prove to be an invaluable resource to the cell cycle and mitosis communities.

PLK1 mediated phosphorylation of CTCF Ser$^{224}$ may also not be a finite event. Coordinated signaling could occur. The Polo Box Domain (PBD) of PLK1 targets the kinase activity by tethering PLK1 to a phosphopeptide (*Lowery et al., 2004*). While CTCF appears to lack an ideal PBD binding site, perhaps PLK1 is recruited to CTCF Ser$^{224}$ by a proximal phosphoprotein (*Elia et al., 2003*). Conversely, what triggers dephosphorylation of CTCF Ser$^{224}$-P? Thus exploring the vicinal events surrounding Ser$^{224}$ phosphorylation could elucidate the G2/M function of this PTM. Furthermore, as CTCF contains 100 S/T/Y amino acids, there exists the possibility that other bona fide phosphorylation events exist. Similarly, with 65 lysines, CTCF acetylation and/or methylation may also occur and orchestrate a much more complex symphony of insulator regulation. Hence by confirming Ser$^{224}$ phosphorylation, we are likely only previewing what could be an expansive CTCF signaling network. While our assays did not elucidate a conspicuous mechanism or functional consequence of CTCF Ser$^{224}$-P, nonetheless its identification and generation of a specific antibody to it can be a useful resource. From a disease perspective, this PTM may also serve as a useful biomarker of dividing cells or prove to be a therapeutic target for competitive or allosteric inhibition, if targeting PLK1 directly proves adverse. Lastly, at least 5000 CTCF binding locations are conserved amongst mammalian genomes and these are often associated with syntenic chromatin domain boundaries (*Schmidt et al., 2012*; *Dixon et al., 2012*; *Vietri Rudan et al., 2015*). As CTCF Ser$^{224}$-P also appears conserved amongst vertebrates, it is an attractive possibility that Ser$^{224}$ phosphorylation along with yet confirmed CTCF PTMs may constitute a conserved code that underlies chromatin organization and gene regulation akin to the histone code (*Jenuwein and Allis, 2001*). Elucidation of such a CTCF code may also reveal insights into how interphase contacts are remembered and restored before and after mitosis. This prospective code may also be bifurcated species-specifically by the degree of conservation of particular CTCF PTMs, potentially exposing mechanisms of speciation. Thus, further unmasking CTCF Ser$^{224}$-P and thoroughly validating the entirety of CTCF PTMs would be invaluable to understanding how the ubiquitous insulator's functions are specified.

## Materials and methods

### Plasmids and cells

The $Ctcf^{S224A}$ and $Ctcf^{S224E}$ mutations were generated by site-directed mutagenesis with Phusion polymerase (NEB) on pcDNA3-CTCF (*Chao et al., 2002*). Mutations were confirmed by DNA sequencing. Recombinant FLAG-CTCF-6xHis and FLAG-GFP-6xHis were made using plasmids pBD39 and pBD40 (*Sun et al., 2013*). The $Ctcf^{S224A}$ and $Ctcf^{S224E}$ mutations were cloned into pFLAG2 with 6xHis C-terminal tags (pBD116 and pBD117) to produce recombinant FLAG-CTCF (S224A)−6xHis and FLAG-CTCF(S224E)−6xHis in *E. coli*. Tet-inducible co-expression of CTCF-3xFLAG and EGFP in mammalian cells was achieved with pBD102 (*Sun et al., 2013*). $Ctcf^{S224A}$ and $Ctcf^{S224E}$ mutations were incorporated into this vector to generate pBD118 and pBD119. To generate Tet-on CTCF, CTCF(S224A) and CTCF(S224E) cells; AseI linearized pBD102, pBD118 and pBD119 were electroporated into SV40T immortalized tetraploid female Rosa26-M2rtTA MEFs or F1-2.1 female mouse ESCs (*Jeon and Lee, 2011*; *Luikenhuis et al., 2001*). Stable transformants were selected with hygromycin. Female mouse ESCs with a monoallelic mutation in *Tsix* (TST-1) were previously described (*Ogawa et al., 2008*). HEK 293 and the rtTA MEFs were passaged in DMEM GlutaMAX 4.5 g/L D-glucose, 110 mg/L sodium pyruvate, 15% FBS, 25 mM HEPES pH 7.5, 0.1 mM MEM non-essential amino acids, 0.1 mM 2-mercaptoethanol, 100 U/mL penicillin, and 100 ug/mL streptomycin at 37C 5% $CO_2$. TST-1 and F1-2.1 ES cells were propagated in the aforementioned media supplemented with LIF to 500 U/mL. Chemical reagents utilized in cell culture conditions are listed in *Supplementary file 1*. All mESCs and MEFs originated from our lab. HEK293 cells were purchased from ATCC. Identity of all lines was confirmed via examination of deep-sequencing datasets of genomic DNA generated in our lab for relevant features (e.g., species, SNPs, presence of transgenes). All cells tested negative for mycoplasma contamination. Quantification of colony size shown in *Figure 6B* was done blinded to cell line identity and doxycycline induction status.

### CTCF immunoprecipitation

Female immortalized rtTA MEFs stably maintaining pBD102 were grown for 72 hr in MEF media supplemented with 100 µg/mL hygromycin with or without 1 µg/mL doxycycline. ~$5\times10^7$ cells from both treatments were manually collected and washed in cold 25 mM HEPES pH 7.5, 150 mM NaCl, 1 mM DTT, 1 mM EDTA, phosphatase inhibitors (10 mM sodium pyrophosphate, 10 mM B-glycerophosphate, 5 mM NaF, 1 mM $Na_3VO_4$) and protease inhibitors (1 mM PMSF, 1 µg/mL pepstatin, and 1 µg/mL leupeptin). Cell pellets were incubated on ice in 5 cell pellet volumes of swelling buffer (10 mM Tris-HCl pH 7.5, 10 mM KCl, 1.5 mM $MgCl_2$, 1 mM DTT, phosphatase and protease inhibitors) and incubated for 10 min. Digitonin was added to the suspension to 20 µg/mL. Nuclei were pelleted at 2000 g 4C. Nuclei were then washed twice in swelling buffer 0.1% Triton X-100 and pelleted. After aspirating the supernatant, the nuclei were frozen at −80C. Frozen nuclei were resuspended in lysis buffer (25 mM HEPES pH 7.5, 2M NaCl, 10 mM EDTA, 0.5% Sarkosyl, 5 mM DTT, phosphatase and protease inhibitors) and kept on ice for one hour. 150–212 µm acid-washed glass beads (Sigma) were washed in 95% EtOH and then lysis buffer. The washed glass beads were then mixed with the cold nuclear lysate by gentle inversion. The glass beads and the viscous phase of the nuclear lysate were pelleted by centrifugation at 2000 g 5 min 4C. The supernatant was passed through a 25G syringe to further reduce the lysate's viscosity and insoluble material was removed by centrifugation at 16000 g 4C. It was then dialyzed 1:250 overnight (3000 mwco) at 4C in dialysis buffer (25 mM HEPES pH 7.5, 300 mM NaCl, 0.1 mM $ZnSO_4$, 1 mM sodium pyrophosphate, 1 mM β-glycerophosphate, 0.1 mM $Na_3VO_4$, 10% glycerol). It was further dialyzed 1:250 another 5 hr at 4C in IP buffer (dialysis buffer with 0.1% Triton X-100). Sepharose 4B (Sigma) was washed with IP buffer with protease inhibitors and used to preclear the dialyzed nuclear extract. Anti-FLAG M2 agarose (Sigma) was washed and blocked with 25 mM HEPES pH 7.5, 150 mM NaCl, 10% Gly-Gly, and protease inhibitors at 4C. The blocked anti-FLAG agarose was washed with IP buffer with protease inhibitors. Nuclear lysate protein concentrations from uninduced and induced cells were equilibrated by Bradford assay. Equal concentrations of lysates were combined with the blocked anti-FLAG agarose and incubated for 5 hr at 4C with rotation. Immunoprecipitations were copiously washed with IP buffer with protease inhibitors. Bound protein was eluted with 25 mM HEPES pH 7.5, 1% SDS and then TCA

precipitated. Precipitates were resuspended and heated in SDS-PAGE sample buffer. Samples were resolved by 7.5% SDS-PAGE followed by Coomassie Blue R-250 staining.

## Mass spectrometry

Immunoprecipitated CTCF was extracted from a Coomassie stained gel. After destaining with 40% ethanol/10% acetic acid, CTCF were reduced with 20 mM DTT (Sigma) for 1 hr at 56°C and then alkylated with 60 mM iodoacetamide (Sigma) for 1 hr at 25°C in the dark. Proteins were then digested with 12.5 ng/µL modified trypsin (Promega) in 50 µL 100 mM ammonium bicarbonate, pH 8.9 at 25°C overnight. Peptides were extracted by incubating the gel pieces with 50% acetonitrile/ 5%formic acid then 100 mM ammonium bicarbonate, repeated twice followed by incubating the gel pieces with 100% acetonitrile then 100 mM ammonium bicarbonate, repeated twice. Each fraction was collected, combined, and reduced to near dryness in a vacuum centrifuge. For the CTCF-CK2 in vitro kinase reaction, CTCF was extracted from a Coomassie stained gel and peptides were extracted in the same manner as described above except 125 ng/µL chymotrypsin (Roche) in 100 mM Tris, 10 mM calcium chloride, pH7.8 was used. Phosphorylated peptides were enriched using NTA-Fe resin as previously described (*Ficarro et al., 2009*). Peptides were eluted with 250 mM sodium phosphate pH 8.9 and then loaded on a precolumn and separated by reverse phase HPLC (Agilent) over a 120 min gradient before nanoelectrospray using an Orbitrap XL mass spectrometer (Thermo). The mass spectrometer was operated in a data-dependent mode. A full scan MS spectrum (injection time of 1000 ms, resolution of 60,000 across 400–2000 *m/z*) was followed by MS/MS for the top 10 precursor ions in each cycle. For the MS/MS on the LTQ (collision-induced dissociation (CID)) was set at 35% energy, injection time of 100 ms, and isolation width 3 *m/z*. Dynamic exclusion for the data-dependent scans was set to 30 s. Raw mass spectral data files (.raw files) were con- verted into. mgf file format using Proteome Discoverer, and then searched against a SwissProt 2015 database containing *Mus musculus* protein sequences (16,724 entries) using Mascot version 2.4.1 (Matrix Science). Mascot search parameters were: mass tolerance for precursor ions was 10 ppm; the fragment ion mass tolerance was 0.8 Da; two missed cleavages of trypsin or chymotrypsin; fixed modification was carbamidomethylation of cysteine; variable modifications were methionine oxida- tion, tyrosine phosphorylation, and serine/threonine phosphorylation. Mascot peptide identifications and phosphorylation site assignment were verified manually with the assistance of CAMV (http:// web.mit.edu/fwhitelab/software.html) (*Curran et al., 2013*).

## Recombinant proteins and In vitro phosphorylation

FLAG-CTCF-6xHis (wild type, S224A, and S224E) and FLAG-GFP-6xHis were made as previously described in Rosetta-Gami B (DE3) *E. coli* (EMD Millipore) using pBD39, pBD40, pBD116, and pBD117 (*Sun et al., 2013*). 2.5 µg FLAG-CTCF-6xHis was in vitro phosphorylated with 250IU Casein Kinase 2 (CK2) (NEB, P6010) and 200 µM ATP (1 µCi [γ$^{32}$P] ATP) for 30 min at 30C. 1 µg FLAG- CTCF-6xHis and 0.5 µg dephosphorylated casein (SignalChem C03-54BN) were also in vitro phos- phorylated with 0.1 µg Polo Like Kinase 1 (PLK1) (Sigma, P-0060) in 5 mM MOPS pH7.2, 2.5 mM β- glycerophosphate, 4 mM MgCl$_2$, 1 mM EGTA, 0.05 mM DTT, and 200 µM ATP (1 µCi [γ$^{32}$P] ATP) for 60 min at 30C. CK2 and PLK1 labeling was detected by SDS-PAGE and autoradiography. For mass spectrometry, FLAG-CTCF-6xHis was phosphorylated with non-radioactive ATP.

## EMSA

DNA EMSA was performed as previously described (*Sun et al., 2013*). dsDNA oligonucleotides to RS14C and RS14C with a mutated CTCF motif were used (*Spencer et al., 2011*). Sequences are listed in *Supplementary file 2*.

## Antibody generation

The phospho-CTCF Ser224 rabbit polyclonal antibody was developed in collaboration with Cell Sig- naling Technology. This antibody was produced by immunizing rabbits with a synthetic phosphopep- tide spanning amino acids 215 to 229 of human CTCF. The antibody was purified by protein A and peptide affinity chromatography.

## Western blot

SDS-PAGE and western blot transfer to PVDF membranes were carried out under standard conditions. Membranes were blocked and blotted in PBS 0.2% Tween-20 4% nonfat dry milk except for blots performed with the CTCF Ser[224]-P antibody where 4% BSA was used in lieu of milk. Membranes were washed with PBS 0.2% Tween-20 and proteins were detected by chemiluminescence using western Lightning Plus-ECL (Perkin Elmer) and film. Antibodies used are listed in *Supplementary file 1*.

## Lambda protein phosphatase assay

SV40T immortalized tetraploid female Rosa26-M2rtTA MEFs were passaged asynchronously, harvested, PBS washed and aliquoted $4 \times 10^6$ cells/tube. Cell pellets were then flash frozen in liquid nitrogen. Pellets were thawed and then sonicated on ice in 1x NEBuffer Pack for Protein Metallo-Phosphatases with 1 mM $MnCl_2$ (New England Biolabs) and freshly added 1 mM PMSF, 1 μg/mL pepstatin and 1 μg/mL leupeptin. $1 \times 10^6$ cell equivalents were treated with 1000U of Lambda Protein Phosphatase (New England Biolabs) at 30C for 15, 30, or 60 min. Lambda Protein Phosphatase was excluded from the negative control sample which was incubated at 30°C for 60 min in buffer supplemented with 10 mM sodium pyrophosphate, 10 mM B-glycerophosphate, 5 mM NaF, 1 mM $Na_3VO_4$. Reactions were stopped with an equal volume of 2x Laemmli sample buffer and heated at 95°C for 4 min. Samples were analyzed by SDS-PAGE and western blot.

## Immunofluorescence

Cells on glass slides were first permeabilized 4 min on ice with CSKT (100 mM NaCl, 300 mM Sucrose, 10 mM PIPES pH 6.8, 3 mM $MgCl_2$, 0.5% Triton X-100) and then fixed in PBS 4% paraformaldehyde for 10 min at room temperature followed by a PBS wash step. Fixed cells were blocked (PBS, 0.2% Tween-20, 4% bovine serum albumin (BSA), 2% normal goat serum (NGS) (Sigma)) for 30 min at room temperature. Antibodies were diluted in blocking buffer. Samples were washed with PBS300.2T (PBS supplemented to 300 mM NaCl, 0.2% Tween-20). Slides were mounted using Vectashield with DAPI (Vector Labs). Slides were imaged with a Nikon Eclipse 90i microscope with Volocity software (Perkin Elmer). Antibodies used are listed in *Supplementary file 1*.

## Mitotic chromosome immunofluoresence

TST-1 mESCs grown under feeder-free conditions were treated for 3 hr in media supplemented with 1 μg/mL colcemid (Thermo Fisher). The mitotic cells were collected by shake off. Cells were then washed in ice cold PBS twice. Then the cells were resuspended in cold cell swelling buffer (CSB) (10 mM Tris-HCl pH 7.5, 10 mM KCl, 1.5 mM $MgCl_2$, 0.1% Tween-20) and incubated on ice for 10 min at an approximate density of $1-2 \times 10^5$ cells/mL. Cell suspension was applied to slides ($2-4 \times 10^4$ cells/slide) by centrifugation at 350 g for 10 min using a Cytospin 3 (Shandon). Cells were extracted by treating the slides for 10 min at 4C in Potassium Chromosome Media (KCM) (120 mM KCl, 20 mM NaCl, 10 mM Tris-HCl pH 7.5, 0.5 mM EDTA, 0.1% Triton X-100). Mitotic chromosomes were blocked with cold KCM 1% bovine serum albumin 1% normal goat serum for 5 min at 4C. Samples were then incubated with primary antibodies diluted in the blocking buffer for one hour at 4C. The mitotic chromosomes were washed for 5 min with cold KCM 3 times. Secondary antibodies were similarly applied and washed from the samples. Slides were then fixed with PBS 4% paraformaldehyde for 5 min at room temperature. Slides were washed once in PBS then sequentially dehydrated in 70%, 80%, 90%, and 100% ethanol. After briefly drying the slides, coverslips were mounted using Vectashield with DAPI (Vector Labs). Slides were imaged with a Nikon Eclipse 90i microscope with Volocity software (Perkin Elmer). Antibodies used are listed in *Supplementary file 1*.

## Mitotic spread karyotyping

F1-2.1 mESCs overexpressing CTCF, S224A or S224E for 6 days were arrested at metaphase with colcemid for 3 hr and harvested by shake off. Harvested cells were then spun down for 5 min at 300 g and resuspended in 0.056M KCl. Cells were incubated in 0.056M KCl for 30 min and spun down at 300 g. Supernatant was removed, leaving 200 μl, and cells gently resuspended. Cells were crosslinked by adding 1 mL of 3:1 Methanol:Glacial acetic acid solution, spun down at 160 g for 10 min at RT. The methanol:acetic acid treatment and centrifugation steps were repeated three times more

for a total of four times. Cells were then resuspended in 200 µl 3:1 methanol:acetic acid solution and incubated at −20°C overnight. The next day, cells were dropped from one meter onto glass slides frozen at −80°C overnight. Slides were mounted using Vectashield with DAPI (Vector Labs) and imaged with a Nikon Eclipse 90i microscope with Volocity software (Perkin Elmer).

## Cell-cycle analysis by flow cytometry

F1-2.1 mESCs overexpressing WT, S224A or S224E CTCF for 6 days were washed with PBS and fixed by slowly mixing in 70% EtOH chilled at −20C. Fixed mESCs were stored at least overnight in −20°C, washed once with PBS, resuspended in PBS and treated with 100 µg/ml of RNase A for 5–10 min at room temperature. Triton X-100 was added to 0.1% and Propidium Iodide added to 20 µg/ml. The sample was then mixed and incubated in the dark for 15 min at room temperature. DNA content profiles were generated via flow cytometry on a FACSAriaIII instrument at the HSCI CRM Flow Cytometry Core. Cell cycle profiles were generated using FlowJo10 (FlowJo, LLC) using the Watson (Pragmatic) model. G1 and G2 coefficients of variation (CVs) were set to 10 and peak locations were adjusted manually to result in the lowest possible root-mean-square deviation (RMSD).

## ChIP-seq

ChIP-seq was performed using undifferentiated $2 \times 10^7$ feeder-free TST-1 cells. Harvested cells were crosslinked for 5 min in 1% formaldehyde. Crosslinking was quenched by the addition of glycine to 125 mM. Fixed cells were PBS washed and cell pellets were frozen at −80C. Approximately $2 \times 107$ cells were used per ChIP. Pellets were resuspended in Lysis buffer 1 (25mM HEPES pH 7.5, 25 mM EDTA, 0.5% SDS). Chromatin was sheared in a Qsonica Q800R for 20 min at 40% power 30 s on 30 s off at 4C to ~150 bp. After centrifuging away debris, 4 volumes of Lysis buffer 2 (25mM HEPES pH 7.5, 187.5 mM NaCl, 1.25% Triton X-100, 0.625% sodium deoxycholate, 6.25% glycerol, 12.5 mM sodium pyrophosphate, 12.5 mM β-glycerophosphate, 6.25 mM NaF, 1.25 mM $Na_3VO_4$) were added with protease inhibitors (1 mM PMSF, 1 µg/mL pepstatin, and 1 µg/mL leupeptin). Sepharose 4B (Sigma-Aldrich) was washed with ChIP Wash Buffer 1 (25mM HEPES pH 7.5, 150 mM NaCl, 1% Triton X-100, 0.5% sodium deoxycholate, 5% glycerol, 5 mM EDTA, 10 mM sodium pyrophosphate, 10 mM β-glycerophosphate, 5 mM NaF, 1 mM $Na_3VO_4$, protease inhibitors) and used to preclear the chromatin samples. 25 µL/IP Protein G Dynabeads (ThermoFisher) were blocked in ChIP Wash Buffer 1 100 µg/mL Acyclovir, 100 µg/mL Zalcitabine, 5% Gly-Gly (Sigma-Aldrich). Beads were then washed with ChIP Wash Buffer 1. Precleared chromatin, 2 µg of anti-CTCF or anti-CTCF Ser$^{224}$-P antibodies (*Supplementary file 1*), and blocked Protein G Dynabeads were incubated together overnight at 4C with rotation. 10% of the precleared chromatin input was also saved. IP samples were washed with ChIP Wash Buffer 1, then with ChIP Wash Buffer 2 (25mM HEPES pH 7.5, 300 mM NaCl, 1% Triton X-100, 0.5% sodium deoxycholate, 5% glycerol, 5 mM EDTA, 10 mM sodium pyrophosphate, 10 mM β-glycerophosphate, 5 mM NaF, 1 mM $Na_3VO_4$, protease inhibitors) and then with TENG (50 mM Tris-HCl pH 8.0, 1 mM EDTA, 50 mM NaCl, 5% glycerol, protease inhibitors). IPs were incubated for 30 min at 37C in TENG with 200 µg/mL RNase A. 10% Input samples were also treated with 200 µg/mL RNase A. IPs were washed with ChIP Wash Buffer 2 and then TENG. IPs were then eluted at 65C in Elution Buffer (50 mM Tris-HCl pH7.5, 10 mM EDTA, 1% SDS). An equal volume of TE (Tris-HCl pH 8.0, 1 mM EDTA) was added to the eluate with 40 µg Proteinase K and incubated for 2 hr at 45C. The input samples were similarly protease treated in parallel. Both IP and input cross-links were reversed at 65C overnight and then phenol/chloroform/isoamyl alcohol extracted. The aqueous phase was further chloroform extracted and then alcohol precipitated. The DNA was suspended in TE. ChIP-seq libraries were generated from both IP and Input DNA samples using the NEBNext ChIP-seq Library Prep Master Mix for Illumina with NEBNext Muliplex Oligos for Illumina (NEB). DNA was quantified with both a Quant-iT PicoGreen dsDNA Assay Kit (Thermo Fisher) and a KAPA Library Quantification Kit (KAPA Biosystems). Libraries were size selected using Agencourt AMPure XP resin (Beckman Coulter) and sizes confirmed on a 2100 Bioanalyzer (Agilent). Sequencing was performed on Illumina HiSeq 2000 instrument (MGH Bioinformatics Core), resulting in approximately 20–25 million paired-end 50 bp reads per sample.

## ChIP-seq analysis

BWA was used to align reads against the mm9 reference genome (*Li and Durbin, 2009*). Alignments were filtered for uniquely mapped reads and duplicates were removed. Input-normalized coverage tracks were generated using SPP (*Kharchenko et al., 2008*). To identify prospective CTCF binding sites, we resolved regions of ChIP-Seq tag enrichment. Tag counts were analyzed in a 1 Kb window over the chromosome length with a 200 bp step and estimated statistical significance of enrichment of ChIP vs input using negative binomial distribution, with the estimate of the mean based on the tag counts in input, and the size parameter(s) selected based on manual inspection of resulting peak calls. Regions of significant enrichment were generated by merging adjacent significantly enriched windows separated by 1 Kb or less. Broad regions of enrichment were called using SPP (*Kharchenko et al., 2008*). Genomic location of the resultant 50,749 CTCF and 912 CTCF Ser[224]-P peaks (z = 6) was categorized using CEAS according to RefGene annotation of introns, exons, and UTRs (*Shin et al., 2009*). CTCF and CTCF Ser[224]-P data were visualized using IGV with ENCODE mESC CTCF ChIP-seq (GSM723015) and RNA-seq (GSM723776) data plotted for reference. Motif search of the ChIP-seq peak datasets (z = 6) and logo generation was performed using MEME-ChIP with default parameters (http://meme-suite.org/) (*Bailey et al., 2009*). To calculate CTCF and CTCF Ser[224]-P coverage over ChIP peaks, bam files of CTCF and CTCF Ser[224]-P ChIP-seq read alignments were converted to bed files. Bedtools coverage was then used to calculate coverage of CTCF and CTCF Ser[224]-P over all CTCF peaks or shared CTCF and CTCF Ser[224]-P peaks (defined as CTCF peaks that overlap with CTCF Ser[224]-P peaks). Linear correlation between CTCF and CTCF Ser[224]-P coverage was calculated using the lm() function in R with default parameters. To calculate number of CTCF motifs falling in CTCF and CTCF Ser[224]-P peaks, bed files of CTCF motifs were downloaded from PWMscan (http://ccg.vital-it.ch/pwmtools) using the mm9 genome and p-value cutoff of $3 \times 10^{-5}$. Number of CTCF motifs falling within each CTCF or CTCF Ser[224]-P peak was then calculated with bedtools coverage using the -counts option. To calculate average conservation of CTCF motifs falling within CTCF and CTCF Ser[224]-P peaks, the mm9 phyloP 30-way conservation tracks (*Pollard et al., 2010*; *Siepel et al., 2005*) were downloaded from the UCSC genome browser (http://hgdownload.cse.ucsc.edu/goldenPath/mm9/phyloP30way) and converted to one genome-wide bigwig track. Average conservation over CTCF motifs intersecting with CTCF and CTCF Ser[224]-P peaks were then calculated using bigWigAverageOverBed. All Wilcoxon rank sum and Student's t-tests indicated in the text were done using the wilcox.test() and t.test() functions in R, respectively, with default parameters. ChIP-seq data have been deposited in GEO (GSE119697).

Previously published cohesin (SMC1 and SMC3) ChIP-seq data were downloaded from GEO (GSE22562) (*Kagey et al., 2010*) and mapped to the mm9 reference genome using bowtie2 with default parameters. Peaks were called using macs2 with default parameters. SMC1 and SMC3 ChIP-seq peaks were then intersected with CTCF and CTCF Ser[224]-P ChIP-seq peaks called above using bedtools with the -u option.

## HYbrid capture Hi-C (Hi-C$^2$)

HYbrid Capture Hi-C (Hi-C$^2$) probes were designed and hybridization to in-situ Hi-C libraries carried out as described previously (*Sanborn et al., 2015*) with 3–5 million F1-2.1 mESCs carrying a WT (clone FCF2), S224A (clone F3), or S224E (clone A2) transgene grown for 2 days with or without 1 µg/mL doxycycline. Probe sets were designed to enrich interactions in the region of interest: chrX:70,370,161–71,832,975 (mm9). Briefly, 120 bp probes were designed around the MboI restriction sites in the regions of interest as previously described (*Sanborn et al., 2015*) and custom synthesized pools of single stranded oligodeoxynucleotides ordered from CustomArray, Inc (Bothell, WA). Single stranded DNA oligos were amplified and biotinylated in a MAXIScript T7 transcription reaction (Ambion). The resulting biotinylated RNA probes were hybridized to 250–300 ng of in situ Hi-C libraries for 24 hr at 65C. DNA hybridized to the RNA probes was pulled down by streptavidin beads (Dynabeads MyOne Streptavidin C1, Life Technologies), washed, and eluted as described (*Sanborn et al., 2015*). The resulting DNA was desalted using a 1X SPRI cleanup and amplified with Illumina primers for 18 cycles to prepare for sequencing.

Hi-C$^2$ libraries were sequenced to a depth of 8–15 million 50 bp paired-end reads. Reads were trimmed using cutadapt with the options –adapter=GATCGATC (MboI ligation junction) and –minimum-length=20. Reads of each pair were individually mapped to the mus and cas reference

genomes using novoalign and merged into non-allelic Hi-C summary files and filtered using HOMER as previously described (*Minajigi et al., 2015*). To avoid computational complexities arising from normalization of sparse, non-enriched regions in the Hi-C contact map, only Hi-C interactions falling within the capture region were analyzed further. For each capture, awk was used to pull out the filtered Hi-C interactions falling within the target region from the HOMER tag directories (awk '{if (($1=='chrX' and and $2 >= 70370161 and and $2 <= 71832975) and and ($6=='chrX' and and $7 >= 70370161 and and $7 <= 71832975)) print}' HiC-filtered-directory/chrX.tags.tsv>chrX.tags.target.tsv). Hi-C contact maps of the capture regions were then generated from these HOMER tags using the 'pre' command of Juicer tools (*Durand et al., 2016a*). The resulting Hi-C contact maps in. hic format were visualized and normalized with the 'Coverage (Sqrt)' option in Juicebox (*Durand et al., 2016b*).

## Insulation and TAD score analysis of Hi-C$^2$ data

Insulation scores were computed for Hi-C$^2$ datasets as previously described (*Crane et al., 2015*, *Giorgetti et al., 2016*) with a few modifications. Briefly, 25 kb resolution 'Coverage (Sqrt)' normalized Hi-C$^2$ interaction matrices were extracted from Juicebox using the 'dump' function of Juicer tools and converted to the cworld matrix format using custom shell and R scripts (described above). Insulation scores were then computed from the cworld matrices using the 'matrix2insulation.pl' cworld script and the parameters '-v –is 125000 –ids 75000 –im sum' and manually normalized using the total number of reads falling in the capture region, rather than total number of reads on the chromosome. TAD scores were then calculated from the insulation scores using the 'insulation2tads. pl' cworld script with default parameters and the Mecp2 TAD boundaries manually defined as the bins chrX: 71,225,000–71,250,000 and chrX: 71,675,000–71,700,000, based on these bins containing CTCF peaks with convergent motifs that aligned visually with the 'box' of the TAD region.

## RNA-seq

RNA from F1-2.1 mESCs carrying a dox-inducible CTCF S224E transgene (clone A2) grown for six days with or without 1 µg/mL doxycycline for 6 days was purified by TRIzol (Thermo Fisher) extraction via manufacturer's instructions. Polyadenylated mRNA was the selected from total cell RNA using oligo(dT) beads (New England BioLabs). Libraries were constructed using the NEBNext Ultra Direction RNA Second Strand Synthesis Module and the NEBNext ChIP-Seq Library Prep Master Mix Set for Illumina (New England BioLabs) and sequenced on a HiSeq 2000, resulting in 50–60 million 50-nt paired end reads per sample. Two biological replicates were performed.

## RNA-seq analysis

RNA-seq reads were trimmed using cutadapt with the options -e 0.2 -q 20 -a AGATCGGAAGAGC -m 20 –overlap 12. Trimmed reads were aligned to the mm9 reference genome using TopHat2 as previously described. PCR duplicates were removed and HOMER used to tabulate unique reads mapping to exons of each gene. DESeq2 was then used to identify differentially expressed (DE) genes. DE genes were counted as those with adjusted p-value<0.01. Only genes with at least one count in one of the four samples were included in the analysis. MA plot was generated using the plotMA() function of DESeq with LFC estimates generated with the lfcShrink() function of DESeq2 with default parameters.

For visualizing RNA-seq reads overlapping codon 224 of CTCF, RNA-seq reads were aligned to the CTCF S224E transgene sequence using the same parameters as above and an index of the CTCF S224E transgene sequence generated using bowtie2 with the default parameters. The resulting bam file of aligned reads was then visualized on IGV.

Overrepresentation analysis of upregulated and downregulated DE genes was done using the PANTHER Classification System (http://www.pantherdb.org). Gene symbols for downregulated or upregulated DE genes used as input lists and the gene symbols of all expressed genes (i.e., all genes used in DESeq2 analysis) were used as the reference list.

Metagene plots of ChIP-seq over DE upregulated, downregulated and all genes were generated via CEAS using bigwig files of reads from CTCF or CTCF Ser[224]-P ChIP-seq (see above).

## Clustal alignment

UniProt indexed CTCF, PLK1, ATM and GSK3B amino acid sequences (*Supplementary file 3*) from mouse (*Mus musculus*), rat (*Rattus norvegicus*), human (*Homo sapiens*), chimpanzee (*Pan troglodytes*), opossum (*Monodelphis domestica*), chicken (*Gallus gallus*), frog (*Xenopus laevis*), zebrafish (*Danio rerio*), sea urchin (*Strongylocentrotus purpuratus*), fruit fly (*Drosophila melanogaster*), and water flea (*Daphnia pulex*) were aligned using Clustal (http://www.clustal.org/clustal2/). The multiple alignments were iterated at each step using the BLOSUM62 substitution matrix with gap opening, extension, and distance penalties set to 10, 0.2, and four respectively. Percent identity matrices and heat maps were generated for the PLK1, ATM and GSK3β alignments.

## Data availability

All sequencing data have been deposited in the National Center for Biotechnology Information GEO repository under accession GSE119697.

## Acknowledgements

We thank Michael Blower for helpful discussion. This work was supported by NSF GRFP grants and a Herchel Smith Fellowship to AJK, and grants from the NIH (R01-GM058839 and the Howard Hughes Medical Institute to J.T.L.

# Additional information

### Competing interests

Jeannie T Lee: Reviewing editor, *eLife,* and is a cofounder and member of the Scientific Advisory Boards of Translate Bio and Fulcrum Therapeutics. Christopher J Fry: Employee of Cell Signaling Technologies. There are no other competing interests to declare. The other authors declare that no competing interests exist.

### Funding

| Funder | Grant reference number | Author |
|---|---|---|
| National Science Foundation | NSF GRFP | Andrea J Kriz |
| Herchel Smith Endowment fund | Herchel Smith Graduate Fellowship | Andrea J Kriz |
| Howard Hughes Medical Institute | | Jeannie T Lee |
| National Institutes of Health | R37-GM58839 | Jeannie T Lee |

The funders had no role in study design, data collection and interpretation, or the decision to submit the work for publication.

### Author contributions

Brian C Del Rosario, Conceptualization, Data curation, Formal analysis, Investigation, Methodology, Writing—original draft, Writing—review and editing; Andrea J Kriz, Data curation, Formal analysis, Investigation, Writing—original draft, Writing—review and editing; Amanda M Del Rosario, Data curation, Formal analysis; Anthony Anselmo, Formal analysis; Christopher J Fry, Resources; Forest M White, Resources, Supervision; Ruslan I Sadreyev, Data curation, Formal analysis, Supervision; Jeannie T Lee, Conceptualization, Formal analysis, Supervision, Funding acquisition, Writing—original draft, Project administration, Writing—review and editing

### Author ORCIDs

Brian C Del Rosario https://orcid.org/0000-0002-9976-1541
Andrea J Kriz http://orcid.org/0000-0002-3395-4498
Jeannie T Lee http://orcid.org/0000-0001-7786-8850

Decision letter and Author response

Decision letter https://doi.org/10.7554/eLife.42341.021
Author response https://doi.org/10.7554/eLife.42341.022

## Additional files

### Supplementary files

• Supplementary file 1. Antibodies and reagents used in this study

DOI: https://doi.org/10.7554/eLife.42341.014

• Supplementary file 2. Oligonucleotides used in this study

DOI: https://doi.org/10.7554/eLife.42341.015

• Supplementary file 3. UniProt accession numbers used in Clustal alignments

DOI: https://doi.org/10.7554/eLife.42341.016

• Transparent reporting form

DOI: https://doi.org/10.7554/eLife.42341.017

### Data availability

High-throughput sequencing data are available in the National Center for Biotechnology Information GEO repository under accession GSE119697

The following dataset was generated:

| Author(s) | Year | Dataset title | Dataset URL | Database and Identifier |
|---|---|---|---|---|
| Del Rosario BC, Kriz AJ | 2019 | High-throughput sequencing data | https://www.ncbi.nlm.nih.gov/geo/query/acc.cgi?acc=GSE119697 | NCBI Gene Expression Omnibus, GSE119697 |

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
