## [Decision Letter]

Thank you for submitting your article "Exploration of CTCF post-translation modifications uncovers Ser^224^ phosphorylation by PLK1 during the G2/M transition" for consideration by *eLife*. Your article has been reviewed by three peer reviewers, and the evaluation has been overseen by Jessica Tyler as the Senior and Reviewing Editor. The following individual involved in review of your submission has agreed to reveal her identity: Nora Engel (Reviewer #2).

The reviewers have discussed the reviews with one another and the Reviewing Editor has drafted this decision to help you prepare a revised submission.

We all considered that this work provides a very useful resource to the community and we only have a few suggestions for improvement, detailed below in the three reviews.

*Reviewer #1:*

Del Rosario et al. addresses the post-translation modification (PTM) of CTCF, a multifunctional protein critical to eukaryotic chromatin architecture and gene expression, and examines how PTM of CTCF may contribute to function. In particular, the authors have performed CTCF mass spectrometry and identified a novel phosphorylation site at Serine 224. A new antibody is generated to recognize CTCF Ser^224^-P. This phosphorylation is shown to be regulated by PLK1 and enriched during G2/M stages of the cell cycle. Using immunofluorescence, the authors identify that CTCF Ser^224^-P is accumulated at certain pericentric regions of the mitotic chromosomes. With CTCF mutants that either obviate serine phosphorylation (S224A) or mimic constitutive phosphorylation (S224E), the authors characterize the functional significance of Ser^224^-P and find that S224A leads to no defect but S224E causes smaller colonies in mouse embryonic stem cells (mESCs). To explore the function of CTCF Ser^224^-P and interpret the observed cell growth defect, the authors find that S224E does not affect CTCF-DNA binding, chromatin architecture, or ploidy, but this mimic of constitutive phosphorylation leads to differential expression of p53 pathway genes in mESCs.

Overall, this paper contains a large amount of work, which is resourceful for studying PTMs and for understanding the multivalent functions of CTCF. The new CTCF Ser^224^-P antibody, its validation and characterization presented in the paper, are valuable resources.

Major comments are:

1) The functional significance of CTCF Ser^224^-P is not clear. It is intriguing that the phospho-obviation mutant (S224A) is normal and the phosphor-mimic mutant (S224E) causes defects in mESC colonies. The data presented here suggest that Ser^224^ phosphorylation seems dispensable for cell survival, but constitutive phosphorylation at Ser^224^ is deleterious to mES cell growth. Since CTCF Ser^224^-P is enriched only at the G2/M transition of the cell cycle and is limited to pericentric regions, the function of Ser^224^-P may be part of the chromatin regulation for mitosis that is multi-layered. However, phosphorylation at Ser^224^ may induce a negative charge and alter the domain conformation, which may change CTCF-mediated gene regulation across the cell cycle. Is it possible that CTCF-protein interactions dependent on this domain surrounding Ser^224^, which appears evolutionarily conserved (Figure 1F), and constitutive phosphorylation at Ser^224^ may stabilize/weaken the protein-protein interactions? The authors should clarify these possibilities.

2) The approach and CTCF mass spectrometry taken in this work fail to detect the previously reported CTCF phosphorylations – Ser^604^, Ser^609^, Ser^610^, Ser^612^ – in CTCF-3xFLAG induced MEFs or through in vitro CK2 phosphorylation of recombinant CTCF. The authors should add details explaining the "technical hurdles" that preclude the identification/validation of the known phosphorylation sites, or address differences in experimental settings that may explain why the earlier studies did not include Ser^224^-P.

3) The phospho-CTCF Ser^224^ antibody shows a low level of cross-reactivity to the unmodified CTCF (Figure 2A and B). Would this contribute to the ChIP-seq profiles showing CTCF Ser^224^-P DNA-binding within a subset of CTCF binding sites, especially during interphase while CTCF Ser^224^-P is not enriched yet? It would be helpful to clarify the chromatin-association feature of CTCF Ser^224^-P, i.e. whether Ser^224^-P is present during interphase and is CTCF Ser^224^-P DNA-bound at defined chromosomal regions beyond the G2/M transition phase.

4) The authors show that both CK2 and PLK1 can phosphorylate CTCF Ser^224^ with in vitro kinase assay combined with mass spectrometry. Given that PLK1 is associated with G2/M transition and is most conserved among vertebrates, PLK1 is proposed as the specific kinase for CTCF Ser^224^-P. However, as the authors indicated, a direct CTCF-PLK1 interaction was not found in either HeLa cells or in CTCF-2xFLAG MEFs, and that the loss of CTCF Ser^224^-P observed with PLK1 inhibition in MEFs may be an indirect effect. Therefore, possible direct evidence supporting Figure 3 "CTCF Ser^224^ is phosyphorylated by PLK1 and prominently labels pericentric chromatin" should include immunofluorescence for PLK1 in combination with CTCF Ser^224^-P on metaphase chromosomes. In addition, is CK2 also involved in regulating CTCF Ser^224^-P phosphorylation? The authors should clarify the data with CK2, with respect to PLK1, as possible kinase for CTCF Ser^224^-P.

*Reviewer #2:*

This manuscript addresses a lingering question in the CTCF field: how can a single protein carry out the multiple functions that are attributed to it, some of which are antagonistic? This is a highly significant study that poses the hypothesis that post-translational modifications may explain the diversity of CTCF roles. Mass-spec experiments identify a novel phosphorylation site at Ser^224^. The authors then proceed to perform an elegant series of experiments to define the function of this phosphorylation event and the enzymes responsible for it. The major finding is that this phospho-CTCF accumulates during the G2/M transition of the cell cycle and that phospho-mutants have a distinct expression profile.

This is a well thought out, logical series of experiments that are designed to progressively answer the questions that arise from each finding. I recommend accepting without revisions.

*Reviewer #3:*

The multi-functional protein CTCF binds to tens of thousands of sites in the genome, serving a variety of functions but it is unclear how this diversity of functions is achieved. One viable idea is that CTCF post-translational modifications contributes to this richness of function, although specific evidence supporting this idea is either sparse or somewhat contradictory. In this manuscript Del Rosario and colleagues identify a novel phosphorylation site at Serine 224 in CTCF. They show that the site is phosphorylated by PLK1. They then attempt to identify the function of this form of CTCF. By generating a specific antibody as well as other reagents, they show this modified form accumulates at the G2/M transition of the cell cycle and is enriched at pericentric heterochromatin. They also performed ChIP-seq on this modified form of CTCF and show that it binds to a subset of CTCF binding sites in the genome. Overexpression of a transgene with a phospho-mimic mutation adversely affected ES cell growth but a series of experiments designed to examine the nature of this phenotype did not reveal anything striking that could account for such a phenotype. In conclusion, the authors demonstrate a new modified form of CTCF but much remains to determine whether this modification contributes to a specific function in the cell.

Figure 3: Treatment with the PLK1 inhibitor, BI 6727. Lower concentration of the inhibitor resulted in a decrease in the number of cell positive for Ser^224^-P whereas higher concentration completely abolished CTCFSer^224^-P staining. What does staining with PCNA and H3S10ph antibodies look like? Are the cells clearly not in G2/M? What about a control antibody for other phosphorylated residues?

Figure 4: ChIP-seq. It would be useful to know more about the location of the CTCF Ser^224^-P peaks. Are they found at high affinity CTCF sites? Widely conserved CTCF sites? Are they found at locations with multiple adjacent CTCF binding sites? Sites where cohesins co-bind?

Figure 8: In the Hi-C experiment using overexpressed CTCF wild-type and mutant transgenes, were the mutant CTCF proteins found at the sub-TAD domain? RNA seq suggests that S224E is only modestly expressed.

---

## [Author Response]

Reviewer #1:[…] Major comments are:1) The functional significance of CTCF Ser^224^-P is not clear. It is intriguing that the phospho-obviation mutant (S224A) is normal and the phosphor-mimic mutant (S224E) causes defects in mESC colonies. The data presented here suggest that Ser^224^ phosphorylation seems dispensable for cell survival, but constitutive phosphorylation at Ser^224^ is deleterious to mES cell growth. Since CTCF Ser^224^-P is enriched only at the G2/M transition of the cell cycle and is limited to pericentric regions, the function of Ser^224^-P may be part of the chromatin regulation for mitosis that is multi-layered. However, phosphorylation at Ser^224^ may induce a negative charge and alter the domain conformation, which may change CTCF-mediated gene regulation across the cell cycle. Is it possible that CTCF-protein interactions dependent on this domain surrounding Ser^224^, which appears evolutionarily conserved (Figure 1F), and constitutive phosphorylation at Ser^224^ may stabilize/weaken the protein-protein interactions? The authors should clarify these possibilities.

Yes, we do believe there is a possibility that PTMs may influence the way in which CTCF regulates genes as well as interactions between CTCF and other proteins. We hope that the resources we provide to the community will encourage exploration of these possibilities. We have modified the Discussion to include several of the scenarios that the reviewer suggests as we also believe they are exceptional starting points for further research.

2) The approach and CTCF mass spectrometry taken in this work fail to detect the previously reported CTCF phosphorylations – Ser^604^, Ser^609^, Ser^610^, Ser^612^ – in CTCF-3xFLAG induced MEFs or through in vitro CK2 phosphorylation of recombinant CTCF. The authors should add details explaining the "technical hurdles" that preclude the identification/validation of the known phosphorylation sites, or address differences in experimental settings that may explain why the earlier studies did not include Ser^224^-P.

We have modified the subsection “Murine CTCF is phosphorylated at a highly conserved position, Ser^224^”, to more thoroughly explain the technical details preventing detection of CTCF phosphorylation at previously reported sites Ser^604^, Ser^609^, Ser^610^, Ser^612^. Namely, these sites reside in a region of CTCF poorly cut by trypsin and chymotrypsin. Therefore, mass spectrometry could not determine which of these sites is phosphorylated. We would also like to note that the previous studies (1-5) did not directly determine, for example through phosphorylation experiments with recombinant proteins, if CK2 was phosphorylating Ser^604^, Ser^609^, Ser^610^, Ser^612^. In addition, the previous mass spectrometry studies which identified these phospho-sites were global studies which did not provide spectra or other further validation of these individual sites, making direct comparison with our study difficult (3-5).

3) The phosphor-CTCF Ser^224^ antibody shows a low level of cross-reactivity to the unmodified CTCF (Figure 2A and B). Would this contribute to the ChIP-seq profiles showing CTCF Ser^224^-P DNA-binding within a subset of CTCF binding sites, especially during interphase while CTCF Ser^224^-P is not enriched yet? It would be helpful to clarify the chromatin-association feature of CTCF Ser^224^-P, i.e. whether Ser^224^-P is present during interphase and is CTCF Ser^224^-P DNA-bound at defined chromosomal regions beyond the G2/M transition phase.

While the cross-reactivity of the CTCF Ser^224^-P antibody with unmodified CTCF could potentially contribute to the ChIP-seq profiles, we do not believe the CTCF Ser^224^-P ChIP-seq is purely a result of this cross-reactivity. If this were the case, we would expect the CTCF Ser^224^-P ChIP-seq to display a low level of binding at all CTCF ChIP peaks, with CTCF Ser^224^-P binding directly correlating to that of CTCF. To test this possibility, we plotted of CTCF Ser^224^-P versus CTCF ChIP-seq coverage at all CTCF peaks (Figure 4D, black dots) and at shared CTCF and CTCF Ser^224^-P ChIP-seq peaks (Figure 4D, red dots). While there was a detectable linear relationship between CTCF and CTCF Ser^224^-P binding signal for both sets of peaks, this was only sufficient to explain half of all CTCF Ser^224^-P binding (R^2^, fraction of variation in CTCF Ser^224^-P binding explained by CTCF binding, for all peaks = 0.48, R^2^ for shared peaks = 0.54). Therefore, we believe at least a subset of CTCF Ser^224^-P ChIP-seq peaks likely represent real binding sites outside of pericentric regions. In addition, as the ChIP-seq was performed in a population of unsynchronized ES cells, which are mostly in interphase (S phase), this would suggest that CTCF Ser^224^-P is present outside of the G2/M phase of the cell cycle as well. We have clarified these possibilities in the second paragraph of the subsection “CTCF Ser^224^-P binds to a subset of CTCF binding sites during interphase”.

4) The authors show that both CK2 and PLK1 can phosphorylate CTCF Ser^224^ with in vitro kinase assay combined with mass spectrometry. Given that PLK1 is associated with G2/M transition and is most conserved among vertebrates, PLK1 is proposed as the specific kinase for CTCF Ser^224^-P. However, as the authors indicated, a direct CTCF-PLK1 interaction was not found in either HeLa cells or in CTCF-2xFLAG MEFs, and that the loss of CTCF Ser^224^-P observed with PLK1 inhibition in MEFs may be an indirect effect. Therefore, possible direct evidence supporting Figure 3 "CTCF Ser^224^ is phosyphorylated by PLK1 and prominently labels pericentric chromatin" should include immunofluorescence for PLK1 in combination with CTCF Ser^224^-P on metaphase chromosomes. In addition, is CK2 also involved in regulating CTCF Ser^224^-P phosphorylation? The authors should clarify the data with CK2, with respect to PLK1, as possible kinase for CTCF Ser^224^-P.

We thank the reviewer for their comments. Although the reviewer is correct that we did not observe direct CTCF-PLK1 interactions in the co-IP experiments, we believe that this negative result does not preclude PLK1 (or any other kinase) from phosphorylating CTCF. Kinase-substrate interactions are highly transient, making their co-IP exceptionally challenging. The Kd’s are probably well above the nM range. Additionally, in order to IP CTCF from the chromatin fraction, CTCF was extracted in 2M salt, anionic detergent, and reducing agent. This combination likely prevents capture of native protein-protein interactions. In general, because of the harsh conditions they are often done in, we believe IP assays are not necessarily demonstrative of bona fide direct protein-protein interactions in the way that in vitro binding or kinase assays are. We have edited the text to clarify this point. Similarly, the metaphase spread immunofluorescence experiments were not intended to directly address whether PLK1 is directly interacting with CTCF because immunofluorescence does not provide the needed resolution to truly determine if two proteins are interacting as enzyme and substrate. As many previous studies have already shown PLK1 to be chromatin associated during mitosis, we expect that the CTCF Ser^224^-P and PLK1 co-IF would show wide overlap, but that PLK1 would also be found widely on chromatin arms outside of CTCF Ser^224^-P bound regions as well; therefore we do not think that this experiment would lend support to PLK1 being the specific kinase of CTCF. We provide direct evidence for PLK1 being the CTCF kinase in mass spectrometry experiments performed with recombinant CTCF and PLK1 (Figure 3F); however similar experiments were also positive for CTCF and CK2 (Figure 1—figure supplement 1). While we still believe that PLK1 is more likely to be the CTCF kinase than CK2 due to both the expression pattern of PLK1 and the appearance of CTCF Ser^224^-P peaking in G2/M, we agree with the reviewer that our experiments did not capture an in vivointeraction with PLK1 or CK2 and have amended the text to more clearly present both as potential CTCF kinases (subsection “PLK1 phosphorylates CTCF Ser^224^”, last paragraph).

Reviewer #3:[…] Figure 3: Treatment with the PLK1 inhibitor, BI 6727. Lower concentration of the inhibitor resulted in a decrease in the number of cell positive for Ser^224^-P whereas higher concentration completely abolished CTCF Ser^224^-P staining. What does staining with PCNA and H3S10ph antibodies look like? Are the cells clearly not in G2/M? What about a control antibody for other phosphorylated residues?

We have added immunofluorescence for cell cycle markers PCNA and H3S10ph after 12h of treatment with the indicated concentrations of BI 6727 (Figure 3C). As shown, percentage of H3S10ph positive cells concomitantly increases with increasing concentration of BI 6727. This indicates that PLK1 inhibition by BI 6727 induces a G2/M block. The increase in G2/M blockage between 100nM and 1000nM BI 6727 may indicate that PLK1 inhibition is not complete at the lower concentration, possibly explaining why there is remaining CTCF Ser224-P in this condition. Agreeing with this, the EC_50_ for BI 6727 in different cell lines ranges from 10-150nM (6-8). We have clarified this point in the text. Unfortunately, we were not able to provide immunofluorescence for a control protein phosphorylated by PLK1, as we were unable to find high-quality antibodies for both such a substrate and its specific phosphorylated form.

Figure 4: ChIP-seq. It would be useful to know more about the location of the CTCFSer^224^-P peaks. Are they found at high affinity CTCF sites? Widely conserved CTCF sites? Are they found at locations with multiple adjacent CTCF binding sites? Sites where cohesins co-bind?

To further characterize the CTCFSer^224^-P peaks we first compared CTCF binding signal at all CTCF ChIP peaks (Figure 4E, red boxplot) versus shared CTCF and CTCFSer^224^-P ChIP peaks (Figure 4E, blue boxplot). Shared CTCF and CTCFSer^224^-P ChIP peaks (representing the vast majority, 95.9% of all CTCFSer^224^-P ChIP peaks) had significantly more CTCF binding signal than CTCF peaks in general (p < 2.2e-16, Wilcoxon test). This suggests that the CTCFSer^224^-P ChIP peaks are indeed found at higher affinity CTCF sites than CTCF ChIP peaks in general.

We then analyzed the nature of CTCF motifs found in CTCF and CTCFSer^224^-P ChIP peaks. While CTCF motifs found in CTCFSer^224^-P ChIP peaks did not tend to be more conserved than motifs found in CTCF ChIP peaks (Figure 4—figure supplement 1E, p = 0.1623, Wilcoxon test), CTCFSer^224^-P ChIP peaks did tend to contain more motifs than CTCF ChIP peaks (Figure 4F, p < 2.2e-16, Wilcoxon test). In other words, while CTCFSer224-P binding sites do not appear to be more conserved than CTCF binding sites in general, they do tend to be found at locations with greater numbers of CTCF binding sites. We also found that shared CTCF and CTCFSer^224^-P ChIP peaks tended to be larger than CTCF ChIP peaks in general (Figure 4—figure supplement 1F), which may in part explain some of these trends.

Finally, we compared our CTCF and CTCFSer^224^-P ChIP-seq peaks with a previously published cohesin ChIP done in mouse embryonic stem cells (9). While 22.8% and 33.3% of our CTCF ChIP-seq peaks overlapped with SMC1 and SMC3 peaks, respectively, 51.6% and 85.1% of our CTCFSer^224^-P ChIP-seq peaks overlapped with SMC1 and SMC3 peaks, respectively. This suggests that CTCFSer^224^-P ChIP-seq peaks also tend to be more found at sites where cohesin co-binds.

We have edited the text to include this further characterization of our CTCFSer^224^-P peak (subsection “CTCF Ser^224^-P binds to a subset of CTCF binding sites during interphase”, last paragraph).

Figure 8: In the Hi-C experiment using overexpressed CTCF wild-type and mutant transgenes, were the mutant CTCF proteins found at the sub-TAD domain? RNA seq suggests that S224E is only modestly expressed.

We thank the reviewer for their comments. To check to what extent the mutant S224E CTCF is bound to the sub-TAD region, we did FLAG ChIP-qPCR in doxycycline inducible S224E-3xFLAG mESCs after 48h of dox induction, the same conditions under which the Hi-C^2^ experiment was performed (Figure 8C). While FLAG binding was detected to be enriched at one of the positive control regions over IgG and the –dox sample or a negative control region (Figure 8C, Cirbp versus Oct4), the level of binding was much lower than that of CTCF. This is consistent with low expression of CTCF S224E versus the endogenous WT CTCF, as suggested by the RNA-seq. We next examined the Mecp2 TAD for CTCF peaks most likely to have the potential to affect chromatin architecture, selecting the CTCF peaks at the TAD borders (Figure 8A, Irak1 and Ikbkg) and well as the peak at the sub-TAD border which is bound by CTCF Ser224-P (Figure 8A, Flna). FLAG binding was modestly enriched over IgG, the -dox sample and a negative control region at the two CTCF binding sites bordering the TAD(Figure 8C, Irak1, Ikbkg versus Oct4), as well as Flna in one replicate, although again the level of binding was much lower than that of CTCF. This suggests that CTCF S224E binding at the Mecp2 TAD is indeed much lower than binding of the endogenous WT CTCF, possibly explaining why we were unable to detect changes in chromatin architecture in the region upon S224E overexpression. We have modified the text to explain this accordingly.

Note that we have also slightly adjusted the TAD borders for the TAD score calculation in 8B to align with the CTCF peaks tested by FLAG-ChIP (Figure 8A, C, Irak1 and Ikbkg).

References:

1) E. M. Klenova et al., Functional phosphorylation sites in the C-terminal region of the multivalent multifunctional transcriptional factor CTCF. Molecular and cellular biology 21, 2221-2234 (2001).

2) A. El-Kady, E. Klenova, Regulation of the transcription factor, CTCF, by phosphorylation with protein kinase CK2. FEBS letters 579, 1424-1434 (2005).

3) K. T. Rigbolt et al., System-wide temporal characterization of the proteome and phosphoproteome of human embryonic stem cell differentiation. Sci Signal 4, rs3 (2011).

4) J. V. Olsen et al., Quantitative phosphoproteomics reveals widespread full phosphorylation site occupancy during mitosis. Sci Signal 3, ra3 (2010).

5) N. Dephoure et al., A quantitative atlas of mitotic phosphorylation. Proc Natl Acad Sci U S A 105, 10762-10767 (2008).

6) D. Rudolph et al., BI 6727, a Polo-like kinase inhibitor with improved pharmacokinetic profile and broad antitumor activity. Clin Cancer Res 15, 3094-3102 (2009).

7) D. Rudolph et al., Efficacy and mechanism of action of volasertib, a potent and selective inhibitor of Polo-like kinases, in preclinical models of acute myeloid leukemia. J Pharmacol Exp Ther 352, 579-589 (2015).

8) R. Gorlick et al., Initial testing (stage 1) of the Polo-like kinase inhibitor volasertib (BI 6727), by the Pediatric Preclinical Testing Program. Pediatr Blood Cancer 61, 158-164 (2014).

9) M. H. Kagey et al., Mediator and cohesin connect gene expression and chromatin architecture. Nature 467, 430-435 (2010).